

# Synchrony in catchment stream colour levels is driven by both local and regional climate

Brian C. Doyle[1,2], Elvira de Eyto[2], Mary Dillane[2], Russell Poole[2], Valerie McCarthy[1], Elizabeth Ryder[2,3], Eleanor Jennings[1]

[1] Dundalk Institute of Technology, Dundalk, Co Louth, Ireland
[2] Marine Institute, Furnace, Co Mayo, Ireland
[3] University College Cork, College Road, Co Cork, Ireland

10  *Correspondence to*: Brian C. Doyle (brian.doyle@dkit.ie)



**Abstract.** Streams draining upland catchments mobilise significant loads of carbon from terrestrial reservoirs to downstream freshwater and marine aquatic ecosystems and ultimately, via a range of biotic and abiotic processes, to the atmosphere. There are increasing concerns over the long-term stability of terrestrial carbon stores in blanket peatland catchments as a result of anthropogenic pressures and climate change. We analysed sub-annual and inter-annual changes in river water colour (a reliable proxy measurement of dissolved organic carbon (DOC)) using six years of weekly data (2011 to 2016) from three contiguous river sub-catchments (Black, Glenamong and Srahrevagh) in a blanket peatland catchment system in western Ireland, assessing differences in catchment characteristics and in the drivers of temporal change. General additive mixed modelling was used to identify the principle environmental drivers controlling changes in colour concentration in the rivers, while wavelet cross correlation analysis was used to identify common frequencies. Although at 130 mg PtCo L$^{-1}$, the colour levels in the Srahrevagh (the subcatchment with lower rainfall and higher forest cover) were almost 50% higher than those from the Black and Glenamong, 95 and 84 mg Pt Co L$^{-1}$ respectively. The  decomposition of the colour datasets revealed similar multi-annual, annual, and event-based (random component) trends, illustrating that environmental drivers operated synchronously at each of these temporal scales, and also spatially within the same catchment. For the Black and its nested Srahrevagh catchment, soil temperature, soil moisture deficit and the weekly North Atlantic Oscillation (NAO) explained 54% and 58% of the deviance in colour respectively. In the Glenamong, which had steeper topography and a higher percentage of peat intersected by streams, soil temperature, log of discharge and the NAO explained 66% of the colour concentrations.  Cross-wavelet time-series analysis between river colour and each environmental driver revealed a significant high common power relationship at an annual time step. Each each relationship however, varied in phase, further highlighting the complexity of the mechanisms driving river colour in the sub-catchments. The estimated mean annual DOC loads for the Black and Glenamong rivers to Lough Feeagh were 15 t C km$^2$ yr$^{-1}$ and 14.7 t C km$^2$ yr$^{-1}$ respectively, although the export values displayed significant inter-annual variation. The results of the study highlight the interaction of catchment conditions and regional meteorological drivers of aquatic carbon export and, therefore the vulnerability of blanket peatland carbon stores to changes in temperature and precipitation. These changes are presently being observed and are predicted to become increasingly extreme and variable.

**Key words.** Carbon transport, dissolved organic carbon, blanket peatland, climate effects



## 1 Introduction

Blanket peat ecosystems occur within a relatively narrow window of climatic conditions, characterised by warmer and wetter conditions, in regions where precipitation exceeds potential evaporation by a ratio of about three to one (Wieder & Vitt 2006). Under such conditions, primary production exceeds decomposition and soil organic matter, and therefore organic carbon (C) accumulates. These ecosystems are a major terrestrial carbon store (Bain et al., 2011). Blanket peatlands are now recognised as being under threat, however, not only from excessive erosion due to anthropogenic pressures (for example, harvesting, burning and grazing) (Renou-Wilson et al., 2011), but also from increases in C loss related to directional climate change (Gallego-Sala & Prentice., 2013). Streams and rivers are the major pathways along which organic C is conveyed from upland peatlands to downstream lakes and oceans. In most studies which have evaluated fluvial losses of both dissolved organic carbon (DOC - mainly from decomposition of peats) and particulate organic carbon (POC – mainly from erosion of peat), DOC has been identified as the more dominant C form, representing between 60% and 88% of the total carbon load (Hope et al. 1997a; Tipping et al. 1997; Ryder et al., 2014). Hope et al. (1997b) concluded that, for British rivers as a whole during 1993, 0.68 Mt C of the fluvial carbon load was in dissolved form, representing 77% of total C export. In the west of Ireland, DOC was estimated to account for 60.5% of the total fluvial C load from the Glenamong sub-catchment (Ryder et al., 2014), which is also used in the current study

Stream DOC concentrations, draining blanket peatland catchments in Ireland, typically show a distinct seasonal pattern, with highest values from late summer to early winter and lowest values in spring (e.g. Ryder et al., 2014). Longer term patterns in DOC concentrations or in proxies for DOC have been linked to year-to-year changes in climate on both local and regional scales. At local scales, temperature affects peat decomposition rates and therefore the availability of DOC, while higher precipitation increases the washout of DOC from soils (Jennings et al., 2010; Ryder et al., 2014). Increases in oxygen availability within peat during droughts can also lead to higher rates of aerobic decomposition (Fenner and Freeman, 2011; Yallop and Clutterbuck, 2009; Mitchell and McDonald, 1992), and may also trigger an enzymatic latch mechanism (Freeman et al., 2001a), where phenolic oxidase activity is switched on in the soil pore waters, reducing the concentration of inhibitory phenolic compounds. At regional scales, DOC concentrations have been shown to be influenced by global weather patterns, for example, certain Canadian lakes have been shown to correlate with climate indices such as the Pacific Decadal Oscillation and the Southern Oscillation Index (Zhang et al., 2010). In Europe, and Ireland in particular, such correlations would be expected to be linked to the North Atlantic Oscillation (NAO). The NAO is a weather phenomenon related to fluctuations in the difference of atmospheric pressure at sea level (SLP) between the Icelandic low and the Azores high (Hurrell et al., 2003). A positive phase of the NAO reflects below-normal atmospheric pressure across Greenland and Iceland and above-normal atmospheric pressure over the central North Atlantic, the eastern United States and Western Europe. A negative phase reflects an opposite pattern of atmospheric pressure anomalies over these regions Ref. High



positive values of this index over northwest Europe are associated with warmer and wetter conditions during the winter and positive index values during the summer are linked with warm, dry and relatively cloud-free periods (Folland et al., 2008). In a 28 year study, Nõges et al (2007) found that water colour (one of the most commonly used proxies for DOC) during spring in Estonian rivers was positively related to the previous winter's North Atlantic Oscillation (NAO) index. A similar positive

relationship between the winter NAO and the total organic carbon (TOC) load was reported over 25 years in Finnish rivers (Arvola et al., 2004).

Increasing fluvial DOC concentrations have been observed in many peat catchments over the last 20 years (Jennings et al., 2010; Erlandsson et al., 2008; Monteith et al., 2007; Worrall and Burt, 2007; Evans et al., 2005; Hongve et al., 2004). While these changes have been attributed in part to recovery from the effects of atmospheric acid deposition on a

regional scale (Erlandsson et al., 2008; Monteith et al., 2007), the trend has also been linked to changes in the key climatic drivers of DOC export. These drivers include the effect of changes in precipitation and snowmelt patterns on flushing rates (Erlandsson et al., 2008; Hongve et al., 2004), and the impact of higher temperatures (Preston et al., 2011; Freeman et al., 2001b) and of drought events (Clark et al., 2005; Jennings et al., 2010) on peat decomposition. There are, however, also studies where DOC concentrations have either decreased (Clair et al., 2008), or no increase has been observed, such as

within certain catchments in the U.K. (Worrall and Burt, 2007), highlighting the need for further insights in the complex relationship between DOC export and climate. Given the close relationship between peat formation and peat decomposition, and climate factors such as temperature, directional climate change is likely to place additional pressures on peatland systems (Clark et al., 2010a; Coll et al., 2014). Observed and projected climate changes for Ireland include higher temperatures throughout the annual cycle, a decrease in the summer water table, and higher winter streamflow (Dwyer,

2012; Nolan, 2015), a combination that has been shown to have the potential to increase fluvial DOC export (Naden et al., 2010). In fact, it has been suggested that modifications to nutrient cycles within peatlands may lead to bogs becoming net emitters of C (Kurbatova et al., 2009). In Ireland, up to 75% of soil carbon storage is in peatlands, much of which is in upland blanket peat soils (Holden and Connolly, 2011, Renou-Wilson et al., 2011). Examining riverine fluxes of carbon from these catchments provides a means to quantify export of C from long-term storage in peatland ecosystems, and to explore the

effects of climatic variables, thus informing future management of peatland systems.

In Europe, Atlantic blanket bogs are found on the western fringes of the continent and are common only in Ireland and Scotland (Sheehy Skeffington and O'Connell, 1998), reflecting the dominant influence of the Atlantic on the local climate in these countries (Coll et al., 2005; Sweeney, 2014). For the Glenamong sub-catchment (one of the sub-catchments in the present study), Ryder et al. (2014) previously reported that soil temperature, river discharge and a dry spring period

explained approximately 60% of the deviance in DOC concentrations over a two-year period. The present study expands on that work, firstly by comparing colour concentrations from three contiguous peat sub-catchments that differ in their catchment characteristics, and secondly by including the role of the regional climatic conditions e.g. the NAO as a possible driver. The principal aims of the current study, using data from the Burrishoole catchment in the west of Ireland were 1. to compare the sub-seasonal, seasonal and multi-annual trends in water colour from rivers in three sub-catchments in a blanket



peatland catchment, 2. to identify the effects of the main climatic drivers and 3. to quantify the inter-annual variability in fluvial export of DOC over the study period.

## 2 Methods

### 2.1 Study Area

#### 2.1.1 Geology and Soils

The Burrishoole catchment (53º 55' N 9º 55' W) is a topographic basin that has been carved into the Nephin Beg mountain range over successive ice-ages and comprises twenty-one lakes of sizes ranging from 0.04 ha to 395 ha and approximately 143 kilometres of interconnecting rivers and streams. Late Precambrian metamorphic rocks and smaller areas of Palaeozoic sandstone and limestone characterise the geology of the catchment (Parker 1977; Long et al., 1992). Rivers and streams on

the western side of the catchment (Glenamong) are generally more acidic, with low buffering capacity (alkalinities in the order of -2.7 to 7.5 mg L$^{-1}$ CaCO$_3$, (Marine Institute, unpublished data) and low aquatic production. Rivers draining the east of the catchment (Black and Srahrevagh) are nearer circumneutral with alkalinities in the order of 15 - 20 mg L$^{-1}$ CaCO$_3$, with consequently higher aquatic productivity. There are also significant till subsoil-deposits throughout the catchment comprising unconsolidated material of lithology reflecting that of its underlying parent bedrock (Kiely et al., 1974). The

overlying soils are predominantly poorly-drained gleys and peaty podsols, with alluvial soils on the valley floors and blanket peatlands covering upland slopes (May and Place, 2005). Land cover in the catchment comprises 52% blanket peat, 15% forestry, with the remaining 33% being made up of discrete parcels of transitional woodland and scrub, natural grasslands and agricultural land (Corine, 2012) (Co-ORdinated INformation on the Environment). Much of the peatland area is commonage, and is used for sheep grazing (Weir, 1996). Vegetation cover on the blanket peats is characterised by *Calluna*

*vulgaris, Molinia caerulea, Schoenus nigricans* and *Scirpus caespitosus* (O'Sullivan, 1993).

#### 2.1.2 Climate

The Burrishoole catchment is located close to the northwest coast of Ireland and experiences a temperate, oceanic climate with mild winters and relatively cool summers. A meteorological station (Newport) has been in operation in the Burrishoole catchment on the shores of Lough Feeagh since 1958. The total annual precipitation at Newport between 2010 and 2016

ranged from a minimum of 1316 mm year$^{-1}$ (2013) to a maximum of 2020 mm year$^{-1}$ (2015), with an average of 1636 mm year$^{-1}$. It is also important to note that spatially, rainfall levels varied spatially across the catchment over the study years from 2,623 mm year$^{-1}$ recorded at an automatic rain gauge in the northwest of the catchment (Namaroon) to 1,508 mm year$^{-1}$ at the south of the catchment (Millrace rain guage) (MI unpublished data). Seasons were defined as follows in the current study: winter = December, January and February; spring = March, April and May; summer = June July and August; autumn

= September, October and November. Maximum summer temperatures in the Burrishoole catchment rarely exceed 20°C,





while minimum winter temperatures are usually between 2°C and 4°C (Marine Institute unpublished data). The maximum air temperature recorded at Newport was 33.9°C in July 2006, while the minimum temperature recorded was -9.0°C in December 2010 (MI unpublished data). The average annual air temperature at Newport meteorological station from 2010 to 2016 was 10.1°C.

### 2.1.3 Catchment characteristics

This study is focussed on three rivers and their subcatchments in the Burrishoole catchment, the Black and the Glenamong, drain directly into Lough Feeagh whilst the third, the Srahrevagh, is nested within the larger Black sub-catchment but differs in terms of forest cover and annual precipitation (Figure 1) (Table 1). The predominant landuse in all three subcatchments are sheep grazing and forestry, however the Black also contains a small proportion of more intensively managed agricultural land. Soils with a peaty, carbon-rich top horizon are common throughout the Burrishoole catchment and blanket peat (technically a soil with peat depth > 40cm) covers approximately 20% of the Burrishoole catchment. Blanket peat has been mapped in all three sub-catchments, with the Srahrevagh containing approximately 5% more peat relative to its area than the other two sub-catchments (Kiely et al., 1974; Gardner and Radford, 1980) (Table 1). The Glenamong has a greater percentage of stream length intersecting blanket peat in comparison to the Black and the Srahrevagh subcatchments (Figure 1 & Table 1). Table 1 also shows the distribution of CORINE (2012) coniferous forestry plantation land cover in the Burrishoole catchment and in the 3 sub-catchments. The CORINE land-cover data shows that 32% of the Srahrevagh sub-catchment contains coniferous plantation planted on peat soil compared to approximately 25% and 17% for the Glenamong and Black respectively (Table 1). The Srahrevagh sub-catchment has the greatest proportion of gentle slopes while the Glenamong is the most mountainous of the three sub-catchments, having the greatest altitude range and containing the greatest proportion of steeper slopes (Table 1). Glenamong stream water is consistently more acidic than the other two sub-catchments; however the remaining stream chemistry metrics between sub-catchments are broadly similar (Table 1). The slope distribution (as percent) for the Burrishoole catchment and each sub-catchment was calculated from a digital elevation model (DEM) at a 10m resolution (Marine Institute Data) using the Spatial Analyst routine in ArcMap 10.3.1.

### 2.2 Water chemistry sampling

Water samples from the rivers were taken at weekly intervals over six years (2011-2016) from the same sampling sites (Figure 1). Colour (mg PtCO L$^{-1}$) was measured within hours of sampling using a HACH Dr 2000 spectrophotometer at 455 nm on water filtered through Whatman GF/C filters (pore size: 1.22 µm).

### 2.3 Meteorological and hydrological measurements

Daily precipitation data, soil temperature data, were available from the Newport met station. Water levels (cm) were recorded every 15 minutes at each site using OTT Hydrometry Orpheus Mini water level loggers. The levels for the Glenamong and Srahrevagh rivers were converted to volume of discharge (m$^3$) using site specific ratings curves that have





been developed using data collected over 20 years (Marine Institute unpublished data). No reliable rating curve was available for the Black river, therefore discharge was estimated using the drainage area ratio method based on the Glenamong discharge (Hirsch, 1979).

## 2.4 Data Analysis

### 2.4.1 Statistical Tests

A Mann-Whitney U test was used to test for statistical differences between colour concentrations in the Glenamong and Black and the Glenamong and Srahrevagh rivers. A Wilcoxon Signed Rank Test was used to test for statistical differences between colour concentrations in the Black and Srahrevagh rivers.

### 2.4.2 Time series decomposition

The time series datasets of weekly colour concentrations in the three rivers were examined using Seasonal Trend Decomposition using Loess (STL) (Cleveland et al; 1990) in R. The STL algorithm decomposes a time series into three separable elements: the trend, the seasonal variation and the residual using an additive model (equation 1). An additive model was preferred over a multiplicative model because no obvious non-stationarity in the time series was observed, i.e. the amplitude of the seasonal cycle remains uniform and does not increase or decrease with the trend in the data (Figure 3A). Variation in time series data was decomposed into a set of constituent elements: overall mean or level ($\alpha$), trend ($T$), seasonal component ($S$), and random noise ($N$) (Chatfield, 1984). For a specific time series for colour concentration in surface waters can be expressed as:

$$Y_{tl} = \alpha_l + T_{tl} + S_{tl} + N_{tl} \qquad (1)$$

where $Y_{tl}$ is the colour concentration at time $t$ in location $l$. The overall mean level ($\alpha$) is site specific and for colour could vary with, for example, the soil, vegetation or land management characteristics of each catchment. Regular variation of seasonal weather, for example, precipitation and temperature, principally drive the $S$ term, and $N$ represents short-term, random events. Any factor that drives the production of colour in surface waters (i.e. DOC production, solubility or transport) over multiple years could drive the trend $T$ (Clark et al., 2010b).

### 2.4.3 General Additive Mixed Models

To identify the main explanatory drivers of colour in the rivers, general additive mixed models (GAMM) with cubic smoothing regression splines and Gaussian distributions were developed using the mgcv package (Wood, 2006). Variance inflation factors (VIFs) less than 3 were used to exclude closely related variables (Montgomery and Peck., 1992, Zuur et al., 2009). All models were tested for violations of the assumptions of homogeneity, independence and normality, and correlation or variance structures included as appropriate. Models were examined for the effects of autocorrelation in residuals by plotting the autocorrelation function (acf) (Venables and Ripley, 2002). All analysis was carried out in R (R



Core Team 2013). The response variable was the weekly colour data from the three sub-catchments. Potential explanatory variables comprising climate and hydrological data were included as continuous variables. The climate variables, measured at the Newport met station, were constructed as follows: the first set was the weekly mean of each climate variable calculated from the sampling week; the second set was the value of each variable measured on the day of sampling. The third

set was constructed by lagging each climate variable by one, two and four weekly time-steps. The climate variables included were: maximum, minimum and mean air temperature (°C), total precipitation (mm), wind speed (ms$^{-1}$) solar radiation (kWh/m$^2$), humidity (%), air pressure (hPa), soil temperature at 5, 10, 30, 50 and 100 cm depths and sun hours (h). Hydrological explanatory variables included river discharge (m$^3$ s$^{-1}$), soil moisture deficit (mm day$^{-1}$) and actual and potential evapotranspiration (mm day$^{-1}$). Soil moisture deficit was calculated using daily precipitation data from the

Newport met station and daily potential evapotranspiration, calculated using Priestly and Taylor (1972). Both weekly and monthly means of the NAO index were downloaded from the National Oceanic and Atmospheric Administration (NOAA, 2017) and used in the statistical analysis. The Standardised Precipitation Index (SPI) was calculated using the R package 'SPEI' (Vicente-Serrano et al., 2010; Beguería et al., 2014) using daily precipitation data from the Newport met station over 21 years (1995 - 2016). The SPI was used to assess relative changes in rainfall and to discover the occurrence of drought

events in the catchment over the study period. The SPI index creates a numeric output on a monthly time step that can be sub-divided into seven categories: extremely wet >2, severely wet 1.5 to 2, moderately wet 1 to 1.5, normal, -1 to 1, moderately dry -1 to -1.5, severely dry -1.5 to -2 and extremely dry < -2. Time series of the possible drivers of water colour, namely soil temperature, SMD, river discharge in the Glenamong and the weekly NAO were also decomposed using the Seasonal Trend Decomposition as described above (Doyle et al., 2018).

### 2.4.4 Cross-wavelet transform analysis

A cross-wavelet transform analysis was carried out to further examine the linkages between each of the explanatory drivers, identified by the GAMM analysis, and water colour in the rivers. Continuous wavelet transforms from pairs of time series are used to construct the cross wavelet transforms, revealing their common power and relative phase in time-frequency space. Regions in the resulting time frequency space that have a consistent phase relationship are suggestive of causality

between the time series (Grinstead et al, 2004). A cross-wavelet power spectrum was calculated from the cross wavelet transform results in order to estimate the covariance between each pair of time series as a function of frequency and the statistical significance was estimated using Monte Carlo methods. The 'biwavelet' package in R (R Core Team, 2017) was used for the bivariate wavelet analyses (Grinstead et al., 2004).

### 2.5 Conversion of colour to DOC concentration, and calculation of carbon export

Water colour (PtCo mg L$^{-1}$) was converted to DOC (mg L$^{-1}$) using a linear model developed between DOC and water colour from the Glenamong River between April 2010 and September 2011(Ryder, 2015). There was a strong linear relationship between colour and DOC was ($r^2$ = 0.88, p ≤ 0.001, n = 366) indicating that water colour measurements are a good proxy for





DOC concentrations in the sub-catchment rivers DOC analysis was carried out using a Sievers 5310C Total Organic Carbon analyser (Range 4 ppb to 50 ppm, accuracy ± 2% or 5ppb). To verify the TOC analyser performance, 10 ppm potassium hydrogen phythalate (KHP) standards were used. Mean annual yield (per km$^2$) was estimated by dividing the mean annual load by the upstream drainage-basin area. The mean load was calculated by multiplying the calculated stream discharge

volume for each week by the weekly DOC concentration and summing the totals to estimate the annual loads.

## 3 Results

### 3.1 Hydrological and meteorological conditions, 2011 - 2016

Weather conditions varied during the six study years, with 2013 was the driest year, with a mean daily precipitation of 3.7 mm day$^{-1}$ and an annual total of 1315 mm year$^{-1}$. The wettest year was 2015 with mean daily precipitation of 5.6 mm day$^{-1}$

and an annual total of 2020 mm year$^{-1}$. The lowest number of rain days (days with >1 mm rainfall) (194) during the six study years was recorded in 2013, compared to 229 rain days recorded for 2015. The maximum air temperature at Newport met station during the period was 29 °C on the 20$^{th}$ of July 2013 and the minimum was −3.1 °C on 3$^{rd}$ January 2011. The warmest summer over the study period was in 2013 with an average mean temperature of 15.7 °C and the coolest summer was in 2011 with an average of 13.3 °C. The coolest winter was in 2014/2015, with an average mean temperature of 5.8 °C

and the warmest winter was in 2011/2012 with an average of 7.2 °C. The driest summer over the study period occurred in 2013 with 258.9 mm accumulated rainfall and the driest winter was also in 2012/2013 with 430.3 mm on accumulated rainfall. The wettest summer was in 2012 with 373.9 mm accumulated and the wettest winter was in 2015/2016 with 744.5 mm accumulated rainfall.

A comparison of monthly precipitation values during the six year study period with precipitation since 1995 at the

Newport met station showed that the first two years, 2011 and 2012, had near normal precipitation totals (SPI of 1 to -1) with some short moderately wet periods (SPI of 1 to 1.5). However, the period between June 2013 and February 2015 was, when compared to the previous 15 years of data, a notable long dry spell, with SPI values ranging from near normal to periods of moderately dry (SPI of -1 to -1.5) to extremely dry values (SPI > -2). Following this predominantly dry period, there was a moderate to extreme wet period from February 2015 to August 2015. Precipitation during the remainder of 2015 and 2016

was mostly near-normal (Figure 2A).

The mean water discharge rates for the three rivers during the study period were 1.89, 0.84 and 0.36 m$^3$ s$^{-1}$ for the Black, Glenamong and Srahrevagh rivers respectively, reflecting the difference in area for the three catchments, as well as the higher precipitation in the Black. The top 10% of discharge in the Black river are > 4.47 m$^3$ s$^{-1}$ and the bottom 10% are < 0.26 m$^3$ s$^{-1}$ (Figure 2B). The highest discharge recorded during the study period for all three rivers occurred on the 5$^{th}$ of

December 2015 with flows of 40.6 m$^3$ s$^{-1}$ recorded for the Black river, 31.8 m$^3$ s$^{-1}$ for the Glenamong and 9.2 m$^3$ s$^{-1}$ for the Srahrevagh. These exceptional flows occurred during Storm Desmond, a 1000-year return event and one of a series of storms




that tracked across the country during a fourteen-week cyclonic episode that began in early November 2015, which brought severe, extensive and protracted flooding to much of Ireland, Scotland and Northern England (Marsh et.al, 2016).

The pattern in soil moisture deficit (SMD) varied considerably over the six years, largely reflecting the varying volumes of precipitation over the catchment each year. The year with the greatest cumulative SMD was 2013 with an average daily deficit of 8.3 mm day$^{-1}$. The cumulative SMD reached a maximum of 66.2 mm day$^{-1}$ in July. The least accumulated deficit occurred in 2015 with an average of 3.9 mm day$^{-1}$, with a maximum of 35.7 mm day$^{-1}$ SMD recorded in July. The maximum daily SMD recorded over the study period was 67.6 mm day$^{-1}$ which occurred in June 2016 however the average daily deficit for that year was 6.4 mm day$^{-1}$ (Figure 2C).

## 3.2 Colour concentrations in the three rivers

The colour concentration showed a strong synchronous annual pattern for all sub-catchments, dipping to a minimum during the winter and peaking in late summer to early autumn (Figure 3A). The Srahrevagh River had the highest colour concentrations, with a median colour concentration of 130 mg Pt Co L$^{-1}$, the Glenamong River had the lowest concentrations with a median of 84 mg Pt Co L$^{-1}$, while the Black River had median values intermediate between these two of 95 mg Pt Co L$^{-1}$. The inter-annual trend in colour concentration (Figure 3B) was also synchronous across all three sub-catchments. There was a peak during the summer of 2012 before it descended to a minimum for all three catchments in the late summer and early autumn of 2013. The trend generally increased from this low-point to the beginning of 2016, with the exception of a minor dip in colour concentration in January 2015. The seasonal patterns were also almost identical for all three sites (Figure 3C) with highest concentrations in late-summer and lowest values in January and February of each year. The Srahrevagh again displayed the greatest range of seasonal variation, and the seasonal variation of the Black and Glenamong were largely similar. The decomposed random component of the colour time-series (Figure 3D) also displayed a broad synchronicity in timing across all three sub-catchments over the six years, indicating that the mechanisms controlling short-term spikes and dips in colour were also synchronous across all three sub-catchments. Similar to the pattern of the seasonal variation, the greatest range of variation in the random component was from the Srahrevagh River (-200 to 169) while the range of variation in the other two rivers were broadly similar (-109 to 107 and -71 to 96 for the Black and Glenamong respectively). Overall, the median colour measurements in the Srahrevagh river were significantly higher than those measured in the Glenamong river (Mann-Whitney U test, W=20317, p<0.01) and Black river (Wilcoxon signed Ranked test, V=892.5, p<0.01). Colour in the Glenamong river was also marginally significantly lower than that measured in the Black river (Mann-Whitney U test, W=37604, p < 0.05).

## 3.3 Drivers of water colour variation

The optimum GAMM model for the colour in the Black River included three smoothers, soil temperature at 100 cm depth, soil moisture deficit, and the weekly mean NAO (Fig. 4). This model explained 54% of the deviance in water colour over the study period (Table 2). Explanatory variables that were measured on the day of sampling resulted in a better model fit than





weekly means for the previous week. Lagging the explanatory variables by one, two and four weeks did not improve the model. The smoother explaining the relationship between soil temperature and colour was linear in the model (edf = 1) and positive, indicating that colour increases with increasing temperature. The smoother describing the relationship between colour and soil moisture deficit was, in contrast, generally negative, indicating that colour concentrations in the river

decreased with increasing SMD, while notably that describing the relationship between NAO and colour indicated that colour decreased in positive phases of the North Atlantic Oscillation. The optimum GAMM for the Srahrevagh River had the same three smoothers as the Black, soil temperature at 100 cm, SMD and the weekly NAO (Figure 4) and the model explained 58% of the deviance in water colour over the study period (Table 2). Again the sub-set of explanatory variables that were measured on the day of sampling provided the optimum model. The smoothers in the Srahrevagh River model also

showed the same patterns in relationship to colour i.e. a positive relationship between colour and soil temperature and a negative relationship between SMD and the weekly NAO.

The optimum GAMM model for colour in the Glenamong River also had three smoothers, but differed in that optimum model included the log of river discharge rather than SMD (Figure 4). The model explained 66% of the deviance in water colour over the study period (Table 2) with again the sub-set of explanatory variables that were measured on the day of

sampling providing the optimum result. The shape of the smoother describing the relationship between colour and discharge indicated that colour concentrations in the river increase to a point and then stabilise at higher discharges. The smoother describing the relationship between the NAO and colour was similar to that found in the other two models and decreased for positive phases of the North Atlantic Oscillation (Figure 4g). The models described above produce the optimum $R^2$ values for the set of explanatory variables chosen. It is important to note that the log of river discharge was also found to be a

significant variable in the models for the Black and Srahrevagh; however, as it was correlated with SMD both explanatory variables could not be used within the same model. When the SMD was used instead of the log of discharge for the Black and Srahrevagh, the revised optimum model variables still included soil temperature and the NAO in both cases with log of discharge, but explained only 48% and 47% of the variance respectively. Conversely, SMD was found to be significant when swapped with the log of river discharge in the Glenamong model, however, the $R^2$ value reduced slightly to 64% from the

optimum value of 66%.

Of note also was the relative importance of the explanatory variables in each of the models. For example, in the optimum model for the Black sub-catchment, out of the total of 54% of the variance explained by the model, soil temperature contributed 34%, SMD contributed 17% and the NAO contributed 3%. For the Srahrevagh, out of the 58% total, soil temperature contributed 40% of the variance, SMD contributed 16% and the NAO contributed 2%. Out of the 66% total

of explained variance for the Glenamong, soil temperature contributed 52%, the log of river discharge contributed 11% and the NAO contributed 3%. The multi-annual trend plots for the NAO, soil temperature, discharge and water colour all had similar patterns that included a distinct dip in the period from late 2012 to mid-2013, and a general upward trend after these low points (Fig. 5). These low-points were sequential for the different variables, with the dip in the NAO occurring in the early winter of 2012, that in soil temperature occurring in early 2013, and that in mean colour concentrations (mean based on



data for all three sites) in mid-summer of 2013. The trends for river discharge, here using the Glenamong as an example, had a less defined low point, which ran from early- to mid-summer 2013 (Figure 5c). The trend in SMD displayed a distinct plateau for each year, with 2013 having the highest levels and 2015 the lowest levels (Figure 5d).

### 3.4 Cross-wavelet power analysis

In the cross-wavelet analysis, there was a significant common power between river colour at the annual (52 week) time scale for all four variables, with additional significant zones occurring intermittently at higher frequencies (*circa* 2-16 weeks) that were most notable for SMD and for the NAO (Fig. 6). For soil temperature, the width of the orientation at the annual time step was relatively consistent with phase arrows that all pointed right, i.e. there was a positive correlation between soil temperature and river colour that was consistent at the annual scale (Fig. 6, top left). The orientation of the phase arrows (i.e.

downward) for stream discharge indicated that river colour was leading river discharge by 90°, indicating they both had seasonal cycles where river colour peaked half an annual cycle before river discharge. The orientation of the phase arrows for the NAO, in contrast, showed a consistent anti-phase or negative correlation with river colour at an annual time step. Notably this weakened between weeks 100 and 150 i.e. during 2013. For SMD (Figure 6 top right) the orientation of the phase arrows at the annual time step, in contrast, showed SMD leading river colour by 90°, therefore, colour peaked half an

annual cycle after a peak in SMD. The areas of common power with river colour at higher frequencies (i.e. at periods of 2 to 16 weeks) were most notable for SMD: these were predominantly negative and occurred during summer periods, when SMD was higher. There were also regular areas of common power at frequencies between 2 and 16 weeks between the NAO and river water colour, however their phase relationship was variable from area to area, with no consistent pattern.

### 3.5 Estimated DOC yields from the sub-catchments

The higher colour concentrations measured in the Srahrevagh throughout the study were apparent in the annual estimated DOC load exported from the sub-catchments. The maximum DOC yield of 38.6 t C km$^2$ year $^{-1}$ estimated for the Srahrevagh catchment during 2015 was almost four times the minimum yield for the study period, 11.6 t C km$^2$ year $^{-1}$ estimated for the Glenamong catchment during 2013. Also notable was the inter-annual variability between study years, 2011 had the greatest total DOC load of 18.5 t C km$^2$ year $^{-1}$ while 2013 was the driest year of the study period has the least total DOC load of 11.8

t C km$^2$ year $^{-1}$. Based on these data, a total of 5898 tons of DOC were estimated to be exported into Lough Feeagh during the six study years.

### 4 Discussion

This study highlighted the dominant influence of local and regional climate on water colour, as a proxy for DOC levels, which explained *circa* 60% of the variability in all three datasets, and the strong synchronicity in these climate signals across

the Burrishoole catchment. It also showed that despite this synchronicity, colour concentrations in one sub-catchment with



greater relative areas of peat soil and forestry (the Srahrevagh) were significantly higher than the other two monitoring sites, a difference that was consistent both seasonally and over the six years of the study. Colour, and therefore DOC, in these headwater rivers originates almost exclusively from the surrounding catchment soils and the quantity of DOC exported during the study was shown to be a function of catchment properties such as the extent of peat within catchments (Hope et al., 1997a), land use (Findlay et al, 2001), local runoff (Dillon and Molot., 2005) vegetation type (Sobek et al., 2007) and the morphology and geology of the catchment landscape (Moore, 1998). Forestry is also known to influence DOC release from soils and it has been observed that both afforestation and forest clear-felling result in increased DOC concentrations and that these increases may continue for several years after the initial event (DeFries & Eshleman., 2004). Spatial analysis, comparing the extent of peat soils in the study catchments, the length of streams intersecting the peat, slope analysis and CORINE land cover in each subcatchment goes some way in explaining the higher levels of colour found in the Srahrevagh. For example, the Srahrevagh contained relatively more mapped blanket peat soils, relatively more gentle slopes (a good indicator for peat soil generation), and relatively more forestry than the other two catchments. An additional factor that may have influenced the variation in colour between the subcatchments could be the distance between a given sampling point and the source of any coloured compounds. Dawson et al. (2002) observed decreases in TOC (both DOC and POC) concentrations in the Upper Hafren (a headwater stream in mid-Wales) downstream from the source that were stated to be related to a decrease in peat depth with altitude, combined with in-stream processing of DOC. A similar process may contribute to the difference in concentration between the upstream Srahrevagh and downstream Black sampling points.

## 4.1 Local climate effects

The local climate effects identified in this study included strong positive linear correlations between colour concentrations in the three rivers and soil temperature. There was also a consistent positive relationship between soil temperature and colour at the annual scale in the cross wavelet analysis. Soil temperature was common to all three GAMMs, and was the dominant explanatory variable, explaining 34% of the variance in the Black, 40% of the variance in the Srahrevagh and 52% of the variance in the Glenamong. Dissolved organic carbon is released by peat soils via decomposition processes that are temperature dependant (Thurman, 1985; Dioumaeva et al, 2002). This temperature effect is complicated however, by an interaction between peat decomposition and the water table level. Peat soils during low water level conditions show increased rates of decomposition, brought about by changes in the oxygen status within the peat and subsequent changes in the microbial community structures within the soil (Mäkiranta et al, 2009). The lowered water table however, reduces the hydrological connection between the source of DOC production and its eventual destination. Increasing temperature, therefore, will increase DOC production in the peat but only if the increase in temperature does not result in a strong draw-down of the water table. The strong relationship displayed between soil temperature and water colour concentrations in the three rivers, and the significant and high common power with river colour at the yearly time scale in the cross-wavelet analysis, indicated that soil temperature was the primary driver of the seasonal pattern in water colour during the study period. These results are consistent with observations of DOC dynamics in many surface waters, where seasonal variation





has been found to be the largest source of DOC variation in catchments with high DOC concentrations (Clark et al. 2010; Ryder et al., 2014). Of note also were the strong similarities in the pattern in multi-annual trends of river colour and soil temperature, particularly in the first four years of the study where the pronounced sequential dip was observed in temperature and then in water colour, indicating as might be expected, that temperature also acts at a multi-annual scale.

The relationship of colour with SMD in the Black and Srahrevagh optimum GAMM models indicated that as soil moisture decreased DOC concentrations, and therefore export, also decreased. The cross-wavelet analysis indicated a significant continuous relationship at a yearly time step where SMD led colour by 90°, that is colour peaked after higher SMDs. More notably there were intermittent, shorter periods during the summer months, where the relationship between the two variables was consistently negative, in line with the GAMM results. The multi-annual trends in decomposed colour

concentrations and SMD over the study period also broadly confirmed this negative relationship, where lower colour concentrations corresponded with higher values in SMD. A drought effect has previously been reported in the Glenamong catchment, where low DOC concentrations in the river were associated with an extremely dry winter and spring during 2010 (Ryder et al, 2014). Hydrology has a direct control on DOC *via* its influence on soil residence time and transfer from soil to stream. For example, the transfer from more lateral drainage routes to vertical pathways with decreasing precipitation would

consequently increase the potential for adsorption of DOC by ion-exchange complexes in mineral soil horizons (Moore and Jackson 1989). There are also indirect effects of hydrology on DOC concentrations such as the availability of water for biogeochemical cycling, biological production and chemical controls on solubility (Clark et al, 2010a). Decreased DOC export has been observed from peat soils during drought conditions where the drawdown of the water table causes the oxidation of organic sulphur to sulphate (Clark et al., 2005; Daniels et al., 2008). This drought-induced soil water

acidification reduces DOC solubility in soil water, and therefore concentrations. However, immediately following periods of dry weather or drought, pronounced increases in DOC concentrations have observed in peatland streams (Watts et al., 2001). Ryder et al. (2014) also reported a significant step-change increase in DOC concentrations in the Glenamong sub-catchment during moderately wet conditions during the summer of 2010 following a very dry spring that year.

River discharge was a significant explanatory variable in the optimum model for the Glenamong, and in alternative

models for the Black and the Srahrevagh rivers, further emphasising the complex effects of hydrology on colour, and therefore DOC concentrations, in the catchment. The reasons why river discharge superseded SMD as an explanatory variable for the Glenamong are likely to be its more westerly location, with higher precipitation, and more mountainous topography. The effect of precipitation has been shown to operate at sub-catchment spatial scales and short temporal scales in the Burrishoole catchment. de Eyto et al. (2016) described the effects of an intense episodic rainfall event on the ecology

of Lough Feeagh, where anomalous high amounts of rain fell in the east side of the catchment (the Black and Srahrevagh) while relatively moderate amounts were recorded in the west (the Glenamong). However, that event was not associated with any increase in water colour, while in contrast, Jennings et al. (2012) described a large increase in DOC concentrations in the Glenamong during increased precipitation in summer 2006. Short-duration high discharge events were also less likely to have been picked up at a weekly sampling regime of the current study. The GAMM smoother for colour concentration in the





Glenamong versus discharge showed that colour increases with increasing discharge to an optimum point where it levels off and even decreases slightly at very high discharges. The cross-wavelet time-series analysis between river colour and river discharge in the Glenamong indicates a significant continuous relationship at a yearly time step with colour concentrations leading discharge by 90°. This high common power relationship is however unlikely to be causal, since colour cannot result

in changes in discharge, but emphasises the co-occurrence of seasonal patterns in both data sets.

## 4.2 Regional climatic effects and the NAO

The current study indicated variable results for the effect of NAO on water colour. There was a negative relationship with the NAO indicated in both the GAMMs, and in the common power at an annual time step in the cross wavelet analysis, but also a lagged positive effect indicated in the plots of the multi-annual trends. The effects of the NAO on the local climate

and lake water temperatures have previously been described for the Burrishoole catchment (Jennings et al., 2000; Blenckner et al., 2007). Jennings et al. (2000) reported that a range of meteorological variables at two lake sites in the west of Ireland (Lough Feeagh, Co Mayo and Lough Leane, Co Kerry) were influenced by the winter NAO, including positive relationships with mean winter air temperature and surface water temperature, mean winter wind speed, and winter rainfall. These relationships were also apparent but to a lesser degree in the following spring and summer. Kiely (1999) also showed that

positive phases of the winter NAO led to increased runoff in Irish rivers. During the construction of the GAMMs, both the weekly and monthly NAO index values were tested in the analysis, with the weekly values consistently producing the optimum results in each of the models, most likely reflecting the proximity of the site to the Atlantic coast and emphasising how rapidly weather associated with the pressure difference reaches the study location. Although the weekly NAO improved the optimum models by only between 2 and 4%, it was significant at the 99.9% level in all of the models. The

smoothers for the weekly NAO in the GAMM's indicated that colour decreased during positive phases of the NAO. This appears to be contrary to the literature for more northern sites whereby DOC has been shown to increase during positive NAO phases, a relationship that has been linked to higher winter precipitation (Arvola et al. 2004; Nõges et al. 2007). However, some studies have also suggested that positive phases of the NAO during the summer are associated with warm and drier rather than warm and wetter conditions over northwest Europe, in particular the UK and much of Scandinavia

(Folland et al. 2008). It is possible that this negative relationship was reflected in the strong effect on SMD on colour in the current study. However, the negative relationship apparent in the cross-wavelet time-series analysis as the annual time step may also merely reflect the fact that .both time series have seasonal patterns, but are not linked by any causal mechanism. Examination of the multi-annual trend of the decomposed NAO values also showed a large and sustained swing in the index to negative values, beginning in late spring of 2012 until spring of 2013, with a subsequent return to positive values.

Negative NAO values during the winter generally correspond to relatively colder and drier conditions, and drier weather was observed throughout 2013, reflected in the SPI Index, beginning during the winter of 2012/2013. Colder conditions are also confirmed by the sharp dip in the multi-annual trend of soil temperature observed during the same winter period. This overall



trend was mirrored in the pattern in water colour but lagged by *circa* six months, suggesting that the NAO actually has a lagged but positive effect on water colour at this time scale. Further analysis with a longer multi-annual dataset would be required to explore the effect of the NAO on water colour and therefore DOC export at this site.

### 4.3 Carbon export from the sub-catchments

The minimum annual total yield from the Burrishoole catchment was 11.8 t C km$^2$ yr$^{-1}$ in 2013 while the maximum was 18.5 t C km$^2$ yr$^{-1}$ in 2011, giving a range of 6.7 t C km$^2$ yr$^{-1}$ during the study period. This considerable variability in total annual yield from the Burrishoole catchment was likely linked to individual sub-catchment characteristics, however, the variability of inter-annual yields was also associated with climate variability as shown by the results presented. For example the driest year of the study period, 2013, displayed the minimum yield. The estimated annual yields reported here are broadly in line with the 9.5 t C km$^2$ yr$^{-1}$ and 13.7 t C km$^2$ yr$^{-1}$ estimated DOC export from the Glenamong sub-catchment in 2010 and 2011 respectively (Ryder et al. 2014) and broadly within the ranges reported in other studies from peatland catchments (Naden et al. 2010; Koehler et al. 2009; Clark et al. 2007)

### 4.4 Conclusions

The results of this study emphasised how colour concentrations, and therefore DOC levels, respond to common climatic drivers which operate at both a local and regional scale, but that they can also vary depending on catchment specific characteristics, even in adjacent or indeed nested catchments. In the Burrishoole catchment, these include area of blanket peat, forestry plantation, and the unique topography within each sub-catchment. Predominantly however, temporal changes in stream colour levels were driven by local changes in soil temperature, hydrology (discharge and/or SMD), and by the NAO, an overarching regional climate pattern. These effects lead directly to variability in DOC export between the sub-catchments that will ultimately affect the carbon budget of the downstream receiving waters of Lough Feeagh. The results presented here serve to further strengthen the well-established link between climate and carbon export from peatland catchments, and the vulnerability of blanket peatlands to climate change.




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



**Figure 1 location of the sub-catchments in the Burrishoole catchment.**





**Figure 2 Panel A: one month Standardised Precipitation Index (SPI) calculated using precipitation data from the Newport climate station (reference period 1995 to 201). Periods where the index is >1 represent moderately wet conditions and periods where the index <-1 indicate moderately dry conditions. Panel B: Black river discharge (grey line - m³ s⁻¹) and mean precipitation (black bars - mm day⁻¹): panel C: cumulative soil moisture deficit per year (mm)(dotted line) and actual soil moisture deficit (mm (grey line)).**





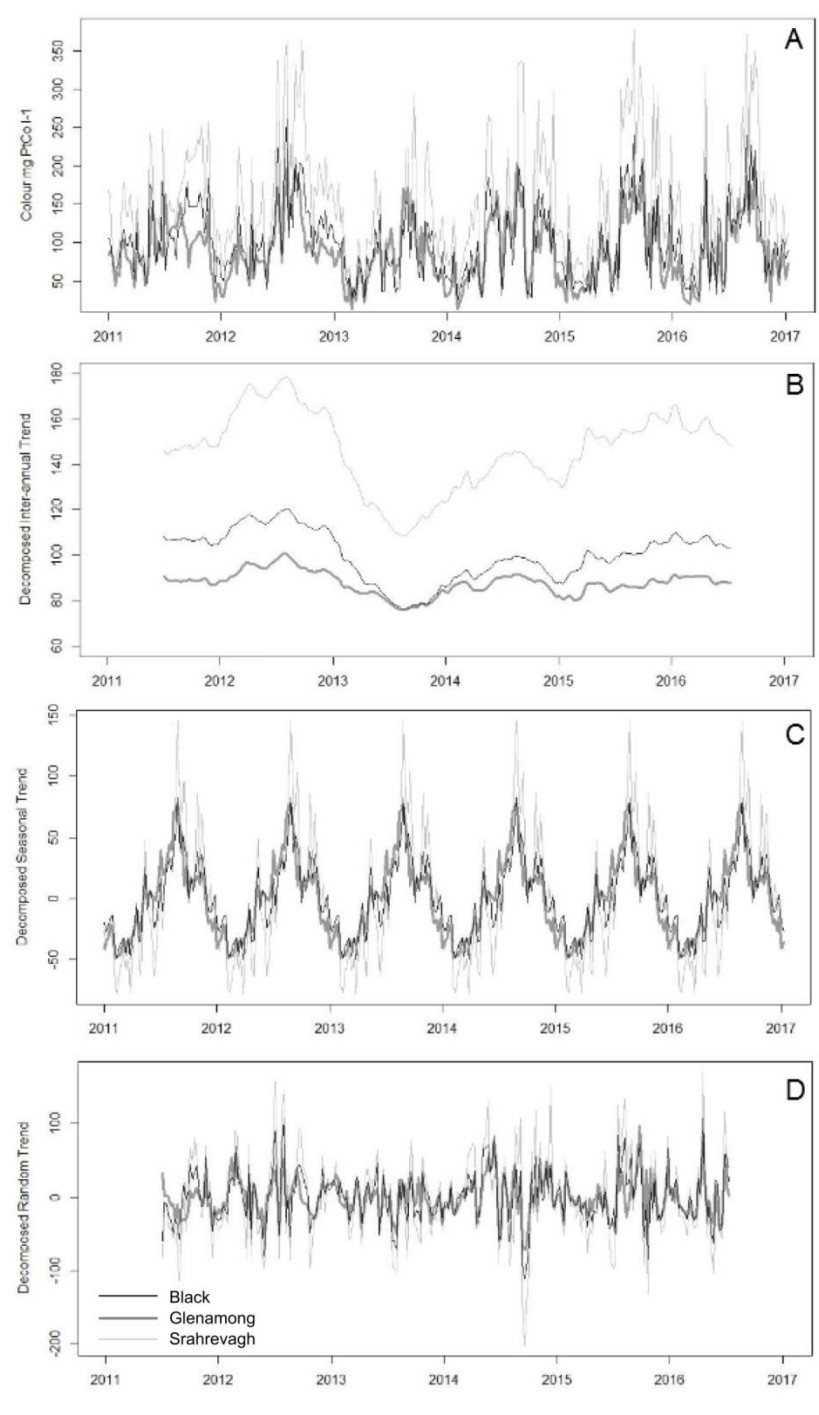

**Figure 3 (A) Time series of colour concentrations (mg PtCo l$^{-1}$) measured weekly in the Black, Glenamong and Srahrevagh rivers from 2011 to 2017 (B) Decomposition of the weekly colour concentrations to the inter-annual trend (C); seasonal component (D) and random component**





**Figure 4 Selected smoothers for the contribution of explanatory variables for the optimal GAMM explaining water colour in each sub-catchment river: Black (a, b, c): a = soil temperature, b = soil moisture deficit, c = NAO; Srahrevagh (d, e, f): d = soil temperature, e =soil moisture deficit, f = weekly NAO; Glenamong (g, h, i): g = soil temperature, h = log of discharge, and i = NAO. The solid line is the smoother and the shaded area shows the 95% confidence bands.**





**Figure 5 (a) Trend of weekly NAO during the study period (b) trend of soil temperature (c) trend of discharge in the Glenamong river (d) trend of SMD and (e) trend of mean colour concentration in the three sub-catchment rivers.**



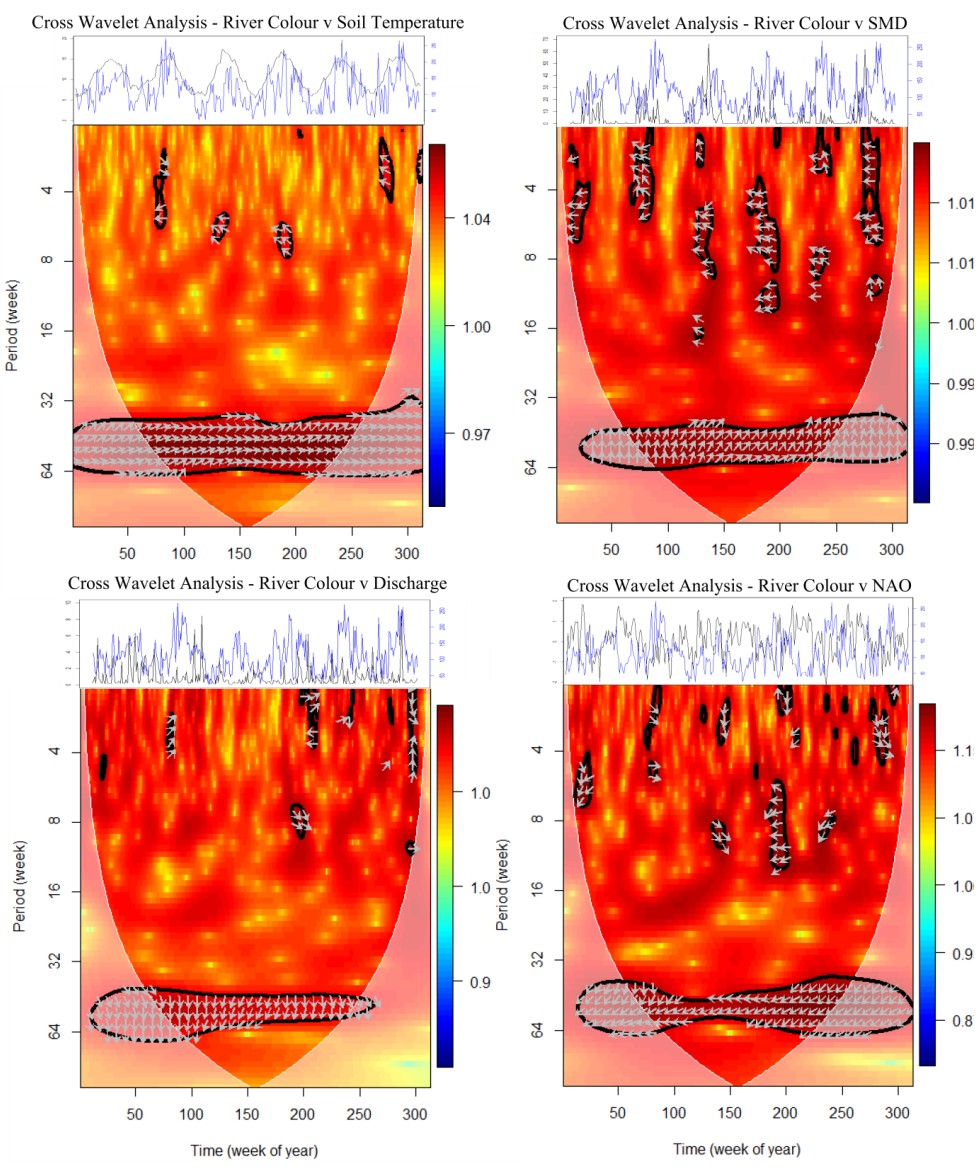

*Figure 6 Cross-wavelet power spectrum of soil temperature (top left, SMD (top right), river discharge in the Glenamong (bottom left) and NAO (bottom right) with river colour. Colour contours represent cross-wavelet power and vectors indicate the relative phase relationship between the two time series (with in-phase pointing right, anti-phase pointing left, or a variable leading or following river colour by 90° pointing straight down/up). The 5% significance level is shown as a thick contour. Pink regions on either end indicate the "cone of influence," where edge effects become important.*



**Table 1 sub-catchment characteristics, climate / hydrology and water chemistry data for the Black, Glenamong and Srahrevagh sub-catchmen**

| Catchment | | Black | Glenamong | Srahrevagh |
|---|---|---|---|---|
| *Characteristics* | | | | |
| Area (km$^2$) | | 48.3 | 17.5 | 4.6 |
| Aspect | | South facing | South-east facing | South-west facing |
| Altitude range (m) | | 8 - 629 | 8 - 710 | 39 - 550 |
| Soils | | Peats, Humic podzols, Regosols and Fluvisols | Peats, Humic podzols, Regosols and Fluvisols | Peats, Humic podzols and Regosols |
| Geology | | Quartzite and schist, also interbedded volcanics, marble, dolomite, and schist | Predominantly Quartzite and schist | Quartzite and schist, also interbedded volcanics, marble, dolomite, and schist |
| Management | | Sheep grazing, Forestry, Grass – Silage | Sheep grazing, Forestry | Sheep grazing, Forestry |
| *Climate / Hydrology 2011 - 2017* | | | | |
| Rainfall (mm yr$^{-1}$) | | 2623 (367) | 2358 (361) | 1853 (204) |
| Mean discharge (m$^3$ s$^{-1}$) | | 5.37 (7.02) | 0.88 (1.26) | 0.42 (0.90) |
| Mean air temperature (°C) | | 10.12 (4.12) | 10.12 (4.12) | 10.12 (4.12) |
| Water temperature range (°C) | | -.05 − 26.0 | -.03 − 26.0 | -.25 − 25.0 |
| *Mean (range) chemistry* | | | | |
| pH range | | 4.0 – 8.0 | 3.5 – 7.3 | 4.5 – 8.0 |
| Colour (mg PtCo l$^{-1}$) | | 15 – 257 | 24 – 211 | 18 – 364 |
| DOC (mg l$^{-1}$) | | 2.3 – 25.8 | 3.6 – 21.5 | 3.2 – 28.7 |
| *Land cover* | | | | |
| Blanket Peat Area % | | 28.7 | 29.7 | 34.4 |
| Stream length (km) | | 87.8 | 37.9 | 12.3 |
| Streams Intersecting peat (km) | | 37.5 | 25.2 | 6.9 |
| Streams Intersecting peat % | | 42.7 | 66.5 | 56.6 |
| CORINE Coniferous Forest % (312) | | 16.6 | 24.9 | 32.2 |
| *Slope (%) distribution in each catchment* | | | | |
| Slope Class (%) | 0 – 10 | *29.2* | *24.4* | *24.2* |
| | 10– 20 | *30.3* | *33.0* | *43.0* |
| | 20 – 30 | *21.2* | *13.9* | *19.5* |
| | 30–50 | *15.7* | *17.3* | *12.8* |
| | 50–100 | *3.6* | *11.4* | *0.5* |




**Table 2, results of Generalised Additive Mixed Models (GAMM) applied to colour in the Black river (top) the Srahrevagh river (middle) and the Glenamong river (bottom) between 2011 and 2017. Stemp100 refers to soil temperature at 100 cm depth, smd refers to soil moisture deficit, dis_glen_log refers to the log of the discharge of the Glenamong river, and nao_we refers to weekly mean values of the NAO index.**

| **Colour – Black River** | *R-sq. (adj) = 0.54* | *Scale est. = 99.816* | *n = 258* |
|---|---|---|---|
| | Estimate | Std. Error | t value | Pr (>|t|) |
| Intercept | 100.30 | 3.41 | 29.40 | < 0.0001 |
| Approximate significance of smooth terms: | edf | Ref.df | F | p-value |
| s(stemp100) | 1.00 | 1.00 | 96.75 | <0.0001 |
| s(smd) | 3.08 | 3.08 | 48.64 | < 0.0001 |
| s(nao_we) | 2.53 | 2.53 | 7.48 | 0.0006 |

| **Colour – Srahrevagh River** | *R-sq.(adj) = 0.58* | *Scale est. = 2809.1* | *n = 261* |
|---|---|---|---|
| | Estimate | Std. Error | t value | Pr (>|t|) |
| Intercept | 145.61 | 4.11 | 35.40 | < 0.0001 |
| Approximate significance of smooth terms: | edf | Ref.df | F | p-value |
| s(stemp100) | 1.00 | 1.00 | 133.08 | < 0.0001 |
| s(smd) | 3.93 | 3.93 | 54.62 | < 0.0001 |
| s(nao_we) | 1.89 | 1.89 | 4.72 | < 0.0067 |

| **Colour – Glenamong River** | *R-sq. (adj) = 0.66* | *Scale est. = 647.71* | *n = 264* |
|---|---|---|---|
| | Estimate | Std. Error | t value | Pr (>|t|) |
| Intercept | 88.52 | 2.27 | 39.06 | < 0.0001 |
| Approximate significance of smooth terms: | edf | Ref.df | F | p-value |
| s(stemp100) | 1.53 | 1.53 | 128.10 | < 0.0001 |
| s(dis_glen_log) | 4.53 | 4.53 | 29.23 | < 0.0001 |
| s(nao_we) | 2.10 | 2.10 | 4.97 | < 0.0081 |





**Table 3 estimated DOC load (t C km² year ⁻¹) from the Black, Srahrevagh and Glenamong catchments between 2011 and 2016. *Totals were calculated as the area-weighted average for the Black and Glenamong sub-catchments only (which are inflows to Lough Feeagh).**

| Catchment | 2011 | 2012 | 2013 | 2014 | 2015 | 2016 |
|---|---|---|---|---|---|---|
| Black | 18.5 | 15.2 | 12.0 | 12.6 | 17.3 | 14.5 |
| Srahrevagh | 28.4 | 25.8 | 21.2 | 35.4 | 38.6 | 18.6 |
| Glenamong | 18.5 | 15.0 | 11.6 | 12.6 | 16.3 | 14.4 |
| Total* | 18.5 | 15.1 | 11.8 | 12.6 | 16.8 | 14.5 |

| Totals by Season: | Black | Glenamong | Srahrevagh | Black | Glenamong | Srahrevagh | Black | Glenamong | Srahrevagh | Black | Glenamong | Srahrevagh | Black | Glenamong | Srahrevagh | Black | Glenamong | Srahrevagh |
|---|---|---|---|---|---|---|---|---|---|---|---|---|---|---|---|---|---|---|
| Winter (D, J, F) | - | - | - | 6.5 | 5.4 | 9.4 | 5.1 | 4.8 | 9.0 | 4.6 | 4.5 | 18.4 | 5.2 | 5.1 | 11.9 | 7.0 | 4.8 | 9.4 |
| Spring (M, A, M) | 3.3 | 3.1 | 4.2 | 1.7 | 1.6 | 2.8 | 3.0 | 3.0 | 5.7 | 2.4 | 2.2 | 6.5 | 3.2 | 2.8 | 5.0 | 2.2 | 2.2 | 2.2 |
| Summer (J, J, A) | 2.7 | 3.1 | 2.4 | 3.5 | 3.8 | 5.6 | 2.2 | 2.3 | 3.1 | 1.4 | 2.1 | 3.4 | 4.0 | 3.8 | 5.9 | 3.1 | 3.4 | 2.2 |
| Autumn (S, O, N) | 7.0 | 7.3 | 11.8 | 4.9 | 4.7 | 8.4 | 2.1 | 2.2 | 3.9 | 3.5 | 3.1 | 5.7 | 3.5 | 5.4 | 16.2 | 4.4 | 4.4 | 8.6 |



**Acknowledgments.** This study was supported by the Marine Institute's Cullen PhD fellowship and core research and development programmes. We would like to thank the local landowners for allowing access to the rivers and providing sites for monitoring stations. With acknowledgement also to the staff at the Marine Institute, Furnace, Co Mayo and Dundalk Institute of Technology, Dundalk, Co Louth without whom this work would not have been possible.

