# Peer review of "Synchrony in catchment stream colour levels is driven by both local and regional climate"

_Biogeosciences, 2018_

## Referee Comment (RC1) · Anonymous Referee #1 · 12 Sep 2018

Reviewing the manuscript entitled "Synchrony in catchment stream colour levels is driven by both local and regional climate" (Doyle et al., submitted to Biogeoscience)

General comments to the authors

The present manuscript analyzes the temporal correlations between climatic and hydrological variables and water color data collected weekly over six years from three blanket peatland catchments in Ireland. Correlations were compared across catchments with different morphological characteristics and across time scales from weeks to years. In addition, inter-annual and seasonal variabilities in dissolved organic carbon (DOC) exports are presented. The correlations were similar in magnitude and direction in all catchments and across all time scales with soil temperature, soil moisture deficit and the NAO index as a key-correlates. Variability in key-correlates and water color

implied variability in DOC exports.

The well-written and well-organized manuscript addresses an important area of research in biogeochemistry / earth system science: how do terrestrial carbon stocks respond to climate change and what is the role of inland waters as a potential export pathway of these stocks? The authors acknowledge the large amount of previous work published on this topic, but in regard to this, should highlight more the knowledge gap and niche that the present manuscript fills. The authors used an interesting approach of cross-wavelet time-series analysis to investigate time scales of magnitudes and directions of correlations. In biogeochemistry, especially peatland biogeochemistry, this analysis is novel and allows new insights in addition to classical correlation analysis. The results and implications of this analysis appear to me a bit drowned among the wealth of other results (which are more straightforward and not very novel, especially in the light of the findings by Ryder et al. 2014). Can the authors highlight the cross-wavelet time-series results more to make a clear stand of how it contributes to new insights? What can we learn from this analysis about the time scales of effects (a good example is given on p. 13, L. 31), what are the problems (as touched upon on p. 15, L. 4-5)?

The two parts of the paper, the analysis of environmental correlates of water color and variability in DOC exports, are interesting in themselves, but what would be even more interesting to many readers was to combine the results of the two analyses: what explains inter-annual and seasonal variability in DOC exports? This is touched upon on p. 16, L. 7-8, but not directly shown. Doesn't the available data allow a more direct analysis of environmental drivers of DOC export (at a seasonal and annual scale)?

The authors highlight both "local" and "regional" climate in the title. To rigorously support this title, wouldn't it be helpful to also explicitly analyze how the regional climate affects the local climate (soil temperature) and to what extent this effect is similar across catchments? If such an analysis would reveal that the regional climate affects local soil temperatures independent of catchment characteristics, it may explain the synchrony

in water color responses to climate across catchments.

To my understanding, there is a mismatch between stated goals, provided analyses and conclusions. In the conclusions, the authors highlight that environmental drivers of water color can differ depending on catchment specific characteristics. However, almost all drivers were the same in magnitude and direction across the different catchments (Fig. 4, and see also p. 10, L. 17-22 and p. 11, L. 13-25). Moreover, the authors only tested for differences in DOC loads (but not drivers) across catchments. This mismatch must be resolved. An analysis of whether drivers differed between catchments would, however, be not very powerful, given that only three catchments were included which, after all were rather similar in key characteristics such as the percentage of streams intersecting peat (Table 1). To strengthen the qualitative discussion of how catchment characteristics modify drivers and magnitudes of DOC loads, it would be valuable to know what range of environmental conditions the studied catchments cover relative to the whole range of Irish blanket peatlands.

Specific comments to the authors

Throughout the manuscript: The authors often use terms such as "controls" and "drivers" (see e.g. p. 2, L. 11). These terms imply mechanistic relationships between environmental drivers and water color. However, the authors used a statistical approach that allows to investigate correlations, not mechanistic links. I suggest to rephrase all terms throughout the manuscript to make clear that relationships were correlative, not mechanistic.

p. 2, L. 1 (Title): Here, the term "climate" is used, but in the abstract (L. 25) the term "meteorological drivers". Please harmonize.

p. 2, L. 4: the term "reservoirs" could be misunderstood, especially by the aquatic biogeochemistry community. Maybe simply use the term "stocks", or "soils" instead of "terrestrial reservoirs"?

p. 2, L. 7-10: This is a very long sentence and hard to digest. I suggest to split it.

p. 2, L. 12: maybe clarify more by adding "in correlations" after "frequencies"?

p. 2, L. 12-14: "Although at 130 mg PtCo L-1, the colour levels in the Srahrevagh (the subcatchment with lower rainfall and higher forest cover) were almost 50% higher than those from the Black and Glenamong, 95 and 84 mg Pt Co L-1 respectively." Why do the authors introduce the sentence with "although"? is it to highlight that the low-rainfall catchment was expected to have clearer water than the other catchments? I would restructure the sentence to get this message better come through.

p. 2, L. 15-16: "illustrating that environmental drivers operated synchronously at each of these temporal scales, and also spatially within the same catchment ": what exactly do the authors want to state here? It reads to me like that environmental drivers were similar across the catchments, but this would contrast to the conclusion that drivers varied depending on catchment-specific characteristics. It would also contrast the statement further down in the abstract (L. 24-25) that "the results of the study highlight the interaction of catchment conditions and regional meteorological drivers". Please clarify.

p. 2, L. 23: why is the term "although" used here? Isn't is enough to simply write that there was inter- annual variation?

p. 2, L. 24: it would be interesting for a wide readership to know whether these inter-annual variations in DOC loads are linked to variability in climate. This remains unclear in the way it is phrased here.

p. 2, L. 24: Can the authors specify what is meant with "interaction of catchment conditions and regional meteorological drivers? what characteristics makes DOC export from a catchment more or less susceptible to environmental drivers? This should be highlighted here, or at least in the conclusions of the manuscript, if supported by the data.

p. 3, L. 3: "warmer and wetter" conditions is relative. Which climate zone is referred to

here?

p. 3, L. 19: what are "year-to-year changes in climate"? climate refers to a period of at least 30 years. I think it is meteorological conditions the authors refer to here.

p. 3. L. 19-25: is an introduction of these enzymatic mechanisms needed? The terms used are quite technical and it seems that it is not relevant for the remainder of the manuscript. I suggest to simply of skip.

p. 3, L. 26: "Canadian lakes have been shown to correlate": what property of these lakes is referred to here?

p. 3, L. 32: which "Ref" is referred to here?

p. 4, L. 25: the authors mention here the implications for future management of peatland systems. Can the authors formulate such implications in the discussion section?

p. 6, L. 11: define "blanket peat" (regarding peat depth) when first mentioned in the manuscript

p. 6, L. 18+20: "gentle" and "steeper" slopes are relative terms. I suggest to refer to absolute numbers here.

p. 6, L. 23: Please add a reference or vendor for the Arcmap program.

p. 6, L. 29: Is it the Newport Met Station that is indicated in Fig. 1? If so, please indicate in the figure and refer to the figure in the text.

p. 6, L. 30: please give the location of the vendor of the water level loggers.

p. 6, L. 31: please report goodness of fit / error measures of the site specific rating curves.

p. 7, L. 6-9: How does this analysis relate to the study aims? Also, I'd appreciate a motivation for the choice of the statistical tests. Was the A Wilcoxon Signed Rank Test used to account for the nestedness of the Black and Srahrevagh rivers?

p. 7, L. 11: please clarify "Loess"

p. 7, L.11: please give a reference for the R program used (move it up from p. 7, L. 29).

p. 7, L. 23- p. 8 L. 19: How were the GAMM models reduced to find the optimum model with three smoothers?

p. 7, L. 25: is the mgcv package an R package? Please indicate. Equation 1: use italics consistently, i.e. even for T_{lt}

p. 8, L. 6: What is the motivation to include wind speed, radiation and humidity in the model? Background / hypotheses for testing these variables are not given in the introduction.

p. 8, L. 12: How did you use NAO in the statistical analysis? As explanatory variables? Please clarify.

p. 8, L. 13: please clarify the abbreviation "SMD"

p. 8, L. 21: what exactly is meant with "to further examine the linkages between each of the explanatory drivers"? I would expect many readers to be unfamiliar with the cross-wavelet transform analysis (including myself) and would appreciate a clarification, in simple words, what the analysis does.

p. 8, L. 27: Please give more details on the Monte Carlo methods used!

p. 8, L. 32: were the residuals of the linear regression between DOC and color homoscedastic?

p. 9, L. 1: add details on the location of the vendor of the DOC analyzer.

p. 9. L 4-5: What is meant with "mean load" and "annual load"? is it the annual mean load referred to here?

p. 9, L. 19: which time period is referred to here? 1995 to ... ?

p. 9, L. 28: is the "top 10%" the 90% percentile?

p. 11, L. 2: what does "edf" stand for?

p. 11, L. 12-25: I very much appreciate the sensitivity analysis, investigating the performance of the GAMM model depending on whether SMD or discharge is included!

p. 11, L. 26-31: How strong was the correlation between NAO, soil temperature and SMD? It comes somehow through in Fig. 5, but some metric describing this correlation might add further valuable context to the relatively low contribution of NAO in addition to the effect of soil temperature and SMD.

p. 13, L. 8: To my understanding, DeFries and Eshleman (2004) only discuss forestry effects on hydrology, not DOC export. Please refer some of the many papers that show increased DOC loads in response to forest clear-felling (e.g. Nieminen 2004, Silva Fennica 38(2); Schelker et al. 2012, GRL). ÂÍ

p. 13, L. 10: how about replacing "goes some way in" by "may help"

p. 13, L. 12: How much is known about the forestry intensity in the catchments? Is the forest clear-cut? In Table 1, only the areal proportion of forest (based on CORINE data) is given, but this does not imply that the forests are managed. Is this the same information that is given in Fig. 1 (symbol code "forestry")? More information on forestry is needed to support the statement that forest clearcutting could explain differences in DOC loads across catchments.

p.13, L. 27-30: Would the interaction with water table fluctuations imply that correlations between soil temperature and water color is low at time scales « 1 year (as apparent from Fig. 6)? If so, I'd suggest to refer to results shown in Fig. 6 here.

p. 14, L. 4: to test this, would it be possible to run the cross-wavelet analysis for time scales longer than 1 year?

p. 15, L. 18: the term "weather" is maybe not optimal here. How about "low pressure

systems" or "cyclones", etc...?

Fig. 1: what do the green-blueish areas mean in the figure? This color code is not explained in the figure legend.

Fig. 1: is the weather station the "Newport" met station? Please indicate.

Fig. 1: please explain the red dot in the map of Ireland.

Fig. 2: explain the abbreviation "SMD" in the figure caption.

Fig. 3: improve the resolution and contrast of the figure

Fig. 3, caption: add "water" in front of "color".

Fig. 3, caption: the letters "B", "C" and "D" appear in the wrong position. Please correct.

Fig. 4, caption: explain the meaning of "s" shown on the y-axis scales.

Fig. 4, caption: explain the abbreviation "SMD"

Fig. 5: I cannot find an explanation in the methods section on how this analysis was done. Please clarify. Details on the trend analysis of water color is given, but not for the environmental driver variables.

Fig. 5: the figure appears stretched along the x-axis in my version. Please modify.

Fig. 5, caption: please indicate the time scale of the trends shown. Is it weekly averages?

Fig. 5, caption: explain the abbreviation "SMD"

Fig. 6: What do the line graphs on top of the contour plots indicate? Also, the tick marks along the axes of these line graphs are hardly visible. Please increase font size and add axis labels.

Fig. 6: Which depth does soil temperature refer to?

Fig. 6, caption: explain the abbreviation "SMD"

Fig. 6, caption: What are edge effects and what is the cone of influence? Please explain here or in the methods section.

Table 1: is the climatological data given here recorded at the Newport met station? Please indicate.

Table 1: "stream length" can differ a lot depending on how / at which spatial resolution it is mapped. how is "stream length" defined? What is the smallest system (e.g. in terms of upslope contributing catchment area) considered here?

Table 2: please explain the abbreviations "edf" and "Ref.df". These values are identical. Why?

Table 3, caption: maybe mention that Lough Feeagh is the lake shown in Fig. 1, or indicate the lake name in Fig. 1?

Table 3: it was not immediately clear to me that the seasonal DOC loads given in the lower part of the table are linked to the years listed in the table header. Maybe explain that in the figure caption?

Table 3: please explain abbreviations D, J, F, ...

Table 3: Shouldn't the sum of the seasonal DOC loads equal the annual DOC loads? This is at least not the case here. Why?

Use consistent abbreviations ("Fig.") for "figure". Some figures are not referred to in the text in the same order as they appear in the figure section. For example, Fig. 3 is referred to before Fig. 2 is referred to.

Typos

p. 2, L. 21: delete one of the "each"

p. 4, L. 10: "trend" should be plural to be consistent with "changes" mentioned before

p. 5, L. 26: delete one of the "spatially"

p. 6, L. 29 replace the first "," by "and", and remove the second ","

p. 7, L. 16: delete one of the "for"

p. 8, L. 23: use "were" instead of "are"

p. 9, L. 1: add a full stop between "rivers" and "doc"

p. 15 , L. 27: remove the full stop between "that" and "both"

Fig. 2: remove "(" after "standardized precipitation index" at the y-axis label of the uppermost panel.

p. 24, L. 4: there is a digit missing in "201"

Table 1 caption: "sub-catchmen" is missing a "t"

---

## Referee Comment (RC2) · Anonymous Referee #2 · 20 Sep 2018

General comments: Doyle et al. present an interesting analysis of patterns in water color variability in three streams partly surrounded by blanket peats in Ireland. By using different statistical techniques they show that temporal patterns in water color are related to local-regional climate (North Atlantic Oscillation, NAO and soil temperature) and hydrology (soil moisture deficit or water discharge). They also estimate DOC export from the water color measurements.

Whereas the scientific questions addressed in this manuscript (ms) are relevant the conclusions are not entirely new. However, the methods used are so far not commonly used in studies of stream dissolved organic matter (DOM), though I suspect their popularity will increase in the future. This study could thus be regarded as a valuable addition to the literature on DOM dynamics in streams.

[Figure]

Specific comments: Though the scientific questions, data and analyses in this ms are relevant and important, the ms could be improved. There are e.g. a few unsubstantiated claims in the discussion, for instance about the relationship between catchment characteristics and water color variability. The authors have not presented any statistical evidence for such relationships, though they seem to have the data necessary. A few simple statistical analyses could thus show if these claims are true or not.

However, I am not sure if the authors really have enough data to show these relationships since they only study three streams. Therefore, I think Doyle et al. should de-emphasize the spatial patterns and concentrate on temporal patterns. The analyses of the temporal patterns are thorough and enough for one paper, and I think removing the speculation about spatial patterns would make for a more coherent paper. Also, the DOC export estimates may be relevant but does not really seem to be an integral part of the study – they seem a little misplaced but I think they could be better integrated into the study, e.g. by relating temporal variability in estimated DOC export to the different potential drivers presented (i.e. NAO, SMD and soil temp).

I would also urge the authors to be more careful about drawing conclusions about DOC dynamics based solely on water color variability. Only a small portion of DOM is actually colored (CDOM; see e.g. Ferrari et al., 1996) and it would be interesting to see an estimate of the differences between water color and DOC concentrations. A discussion about the uncertainties related to the assumption of water color equaling DOC would also be in order. For instance, from the information presented in the ms it seems like there is a larger difference in range in water color in Srahrevagh than in DOC. This has been observed elsewhere and has been hypothesized to be due to changes in iron concentrations (since iron also influence water color; see e.g. Kritzberg & Ekström, 2012).

I would also like to see an analysis of the covariation among the different "independent" variables that the authors use to explain patterns in water color, e.g. NAO, soil temp and SMD. I guess some of these variables co-vary and it would be interesting as a

reader to see HOW they co-vary (could fit in an appendix).

The text is well organized but sometimes feels a bit sloppy and needs to be overhauled – there are e.g. some strange wording and very dense text at times. For instance, some sections of the text needs to be divided into paragraphs. Also, the authors need to use a consistent terminology – as is the ms is sometimes confusing due to varying terminology. For instance, you need to be clear about if DOC concentrations or DOC export is the variable under study. In addition, sometimes the authors use "DOC yield" and sometimes "DOC load" but it is not clear if these terms refer to the same thing or are different concepts (are both concepts needed?). Also, since you are using color as a proxy, you are really only getting information about the colored part of DOM, i.e. CDOM.

The introduction and discussion could be better connected to the literature. I have several suggestions for references and some additional discussion topics (see technical comments).

There are sometimes mismatching information. For instance, in Table 2 the authors indicate that data from 2011 to 2017 was used but in the main text I got the impression that data were from 2011 to 2016. Also, there is some inconsistent use of acronyms in the text compared to figures, and inconsistent use of letters to indicate different panels in figures (sometimes upper-case letters, sometimes lower-case letters)

To summarize, the ms is worth publication but only after a thorough revision, including an overhaul of the text, a discussion about uncertainties relating to the actual differences between water color and DOC, perhaps removing the statistical patterns and a better integration of the DOC export estimates.

Technical comments:

Abstract

Line 5: well, all of this carbon is not transferred to the atmosphere since some of it may

be stored in long-term deposits such as lake or ocean sediments.

Line 10: Temporal change in what?

Line 12: I guess unit should be mg Pt Co L-1.

Line 12-14: I find this sentence a little odd; I expect something to follow the "Although ...". Ok, so the colour concentration was higher in Srahrevagh, but why "although"?

Line 17-18: Does these numbers (54% and 58%) refer to 1) soil temperature + soil moisture deficit and 2) NAO or to 1) soil temperature and 2) soil moisture and NAO. There are only two numbers but three variables making this sentence unclear. In the next sentence you refer to the combined effect of three variables; why do you not do that here?

Line 21: remove one "each"

Line 23: You use different number of digits here. Also, I guess these numbers are per km2. So should it be 15.0 and 14.7 t C km-2 yr-1? And why do you only report load for two of the three catchments?

Line 25: but the analysis you refer to above relate to concentrations. At least in the abstract, you do not mention what controls the C export.

Introduction

P 3 line 5: What do you mean by "...primary production exceeds ... soil organic matter"?

P 3 line 11-12: I suggest you remove the assumed sources of DOC and POC in the parentheses, partly because the sentence is general in form (for instance, not all DOC comes from peat degradation in many catchments) and partly because I am not aware of studies that clearly identify the primary source(s) of DOC or pathway of DOC formation. Previous studies in the UK have e.g. shown that DOC generally is of recent origin, i.e. post 1950s (see e.g. Evans et al., 2007 or Billett et al., 2007). This does not mean that old peat is NOT decomposed but that most of the DOC comes from younger

sources (could, however, be young peat!).

P3 line 12: remove "more" before "dominant"

P 3 line 12-16: You need to be careful with the terminology here. Do you e.g. mean that 60 and 88% of total carbon load, i.e. including DOC, POC, DIC and PIC, were DOC? Or do you mean that 60-88% of total organic C was DOC? Whereas DOC may dominate in many areas, this is often not true where there are large portions of calcareous bedrock within the catchment. Thus, you need to clarify if you mean total C (i.e. including inorganic C) or total organic C.

P3 line 20: yes, but not only decomposition but temperature will also affect other potential DOC forming processes, e.g. root exudates from primary producers

P 3 line 26: I guess it is the DOC in the lakes that correlate with climate indices, not the lakes themselves, or?

P3 line 29: remove the acronym SLP – it is not being used anywhere else in the manuscript and thus superfluous.

P 3 line 32: What should Ref be?

P4 line 14-16: This was also found by Winterdahl et al. (2014) where TOC was increasing in about half of 130 streams, but without any clear geographical patterns. Also worth mentioning is that some authors claim that DOC concentrations have stopped increasing or are actually decreasing (Worrall et al., 2017) whereas others have pointed to methodological differences among studies that limit interpretations of potential trends (Filella & Rodriguez-Murillo, 2014).

P4 line 16: But the studies referred above did not study DOC export but DOC concentrations. Once again you need to make clear if it is export or concentrations that are the focus of study. The export is to a large extent controlled by water discharge, and thus ultimately by the difference between precipitation and evapotranspiration.

P4 line 21-22: This sentence seems out of topic – you have not discussed effects of changes in nutrient cycles before and Kurbatova et al. studied Russian bogs which I suspect behave quite differently compared to the blanket peats on the British Islands (in terms of e.g. hydrology and topography).

P4 line 28-29: Repetition. You have already mentioned that this is one of your study catchments.

P4 line 32: change to "...climatic conditions, e.g. the NAO, as a possible..."

P4-P5 line 33 ff: The aims need clarification. First you mention the Burrishoole catchment but later you write "water colour from rivers in three sub-catchments in a blanket peatland catchment" – why not specify that this is the Burrishoole catchment? Also, part 2 need to be specified; the effects of main climatic drivers on what?

P5 line 6-8: It would be interesting to know the area of the entire Burrishoole catchment.

P5 line 18: You can remove the (Co-ORdinated INformation on the Environment) but write CORINE in capital letters (as you do in the reference list).

P5 line 24: why do you report precipitation only for 2010-2016 when you obviously have a longer time series of precipitation from the area? The mean precipitation for 2010-2016 is reported in the results anyway.

P5 line 26: you repeat "spatially" here – remove one

P5 line 27: Above you did not use a thousand separator (,) but here you do. You need to be consistent and comply with the format of the journal.

P5 lines 26-28: Are these precipitation numbers from the same year? Or are they annual means? That is not clear now. I think you need to show the spatial variability better because as it is now, it is not clear how these observations differ from Newport (besides that the numbers are a bit different). You could perhaps show how large the spatial differences are on average among years (including all three stations with precip.

[Figure]

data).

P6 line 8: I guess this should be (Figure 1 and Table 1)

P6 line 16: change to "three sub-catchments"

P6 line 23: Change to "10 m resolution"

P6 lines 26-27: What was the precision of this instrument? If you have data on accuracy, that would also be relevant to report here.

P6 line 29: "Daily precipitation and soil temperature data..."

P6 line 30- ff: How was the rating curve calibrated? I.e. what methods were used to construct the rating curve? I also think you should report the accuracy of this rating curve, e.g. with an R2.

P7 line 6-8: Why two different tests?

P7 line 16-17: This sentence needs to be rephrased. Should the second "for" be removed?

P7 line 21: I know many authors equal colored DOM and DOC but since only a (small) fraction of DOM actually is colored (see e.g. Ferrari et al., 1996) you might want to refer to CDOM here (and at other places where you use color to draw conclusions about DOC) instead,

P7 line 23: General additive mixed models – this section is a little dense. Could you please divide it into a few paragraphs?

P8 line 6: Unit should be (m s-1).

P8 line 7: So the humidity here should actually be "relative humidity"

P8 line 8: How was actual evaporation estimated?

P8 line 18: I guess SMD is soil moisture deficit but the acronym has not been defined.

P8 line 30: This should be rephrased. Water color was not converted but DOC was estimated from water color. I think you need to be clearer about this throughout the manuscript.

P9 line 1: There should be a . after "sub-catchment rivers".

P9 line 2: Is this really the accuracy? Or is it the precision? If this is the accuracy, what then is the precision? Also, on this line it should be "5 ppb".

P9 line 3: New paragraph before "Mean annual yield. . ."

P9 line 4-5: This sentence should be moved so that it precedes the previous sentence, i.e. first this sentence (starting with "The mean annual load. . .") and then the sentence starting with "Mean annual yield. . ."

P9 line 5: I am a bit confused here, but I think you mean the estimated DOC here, right?

P9 line 8: Should it be ". . .year, with 2013 being the driest year, with. . ." or something similar?

P9 line 16-17: Do you really have the precision to report these numbers with one decimal? Above you did not use a decimal and I think you should be consistent here (also, how many decimals are realistic based on your measurement equipment?).

P9 line 28: This would be easier to see if you also report the specific discharge in e.g. mm/d.

P10 line 5: But the cumulative SMD should have unit mm (only), right?

P10 line 20-22: This "random component" does not seem to be entirely random, at least not from what I can tell from figure 3D. How does the autocorrelation of this random component look like? Would it be possible to subtract even more information from this time series (though I have no idea how to do that)?

P10 line 25-28: This information seems misplaced. Why not combine this with the text in the beginning of this section where you also refer to which stream having the highest concentrations?

P11 line 13: "...the optimal model..."?

P11 line 31: Not sure if I agree about discharge here. Based on figure 5, NAO, soil temp and water color seem similar but the increasing trend in discharge starts more than a year after the increase in NAO.

P12 line 20-25: This section is unclear. Is the second set of numbers reported (i.e. 18.5 and 11.8 t C km2 yr-1) averages among all the study streams? It is not clear how these differ from the first set of numbers (which apparently were for individual streams in individual years). Everything becomes clear when looking at Table 3 but it should be clear from the text as well. Also, sometimes you use the term yield and sometimes load – do these mean different things here?

P12 line 24: This sentence is a bit confusing. I think you should change this to "...while 2013 had the least total DOC load...". You have already reported that 2013 was the driest year – there is no need to reiterate that here.

P12 line 28-29: You should rephrase this sentence. It is unclear, probably because of the misplaced modifier "which" that refer to "DOC levels" or possible to "water colour" in this case. I also think you should change the statement "explained circa 60%" to "explained between 54 and 66 %".

P13 line 2-3: Though I suspect you are right, do you have data to confirm this statement? Do you e.g. have fluorescence data that indicate that DOC primarily is of terrestrial origin? If not, I think you should be more careful and write something like "...probably originates primarily from the surrounding catchment...".

P13 line 3-6: If I understand this sentence correctly you claim that you have shown that the DOC export from the different study catchments in your study are related to

catchment properties, land use, runoff etc. But this is incorrect, you have not shown this. There are no data that show these relationships.

P13 line 8-10: But you do not present such an analysis – you only have three sites, so the statistics will be a little shaky, but can you find any of these relationships that you mention? Figures in an appendix could be enough to show if there are any relationship between water color and e.g. the extent of peat soils in the catchment.

P13 line 14-17: Perhaps, but other studies have not found any clear downstream patterns in DOC concentration (see e.g. Temnerud & Bishop, 2005 and Creed et al., 2015) or clear signs of DOC degradation as water moves downstream in a stream network (see e.g. Winterdahl et al., 2016).

P13 line 21-23: Is it necessary to reiterate the results here?

P13 line 24: Wouldn't Christ and David (1996) and Neff and Hooper (2002) be more relevant references here since they have actually looked at the relationship between temperature and DOC "production/leaching".

P13 line 28-30: You touch upon this but it could perhaps be clarified. You need to think of what you, and most other scientists in this business, refer to as "DOC production" as two different processes (if we simplify everything and ignore e.g. sorption dynamics, solution/dissolution due to changes in water chemistry etc.): 1) the actual DOC production, i.e. some process that forms DOC (could be e.g. exudation of organic molecules through roots or microbial degradation of solid organic matter), and 2) transport of DOC along active flow pathways in the soil. Process 1) could be active as long as there is water in the soil, even if this water is not moving. Process 2) only happens when the water is actually moving. That is, you could have an area with stagnant soil water where DOC production (process 1) forms a "stock" of DOC that is transported to a nearby surface water body as soon as the flow pathways are activated.

P13-14 line 33-1: But this is not generally the case for DOC, see e.g. data from about

130 streams in Winterdahl et al. (2014) where there is no relationship between seasonality and DOC concentration.

P14 line 6: As I understand this, you mean that since concentrations decrease, the export will also decrease. In this case, this is probably true since if soil moisture decreases, stream discharge will also likely decrease. But generally, you can have decreasing concentrations but increasing export if discharge increases. Since discharge on event scales can vary by several orders of magnitude whereas concentrations seldom vary by more than a factor 10, discharge often control the export dynamics, at least on short time scales. Therefore, I think you should remove "…and therefore export…" here.

P14 line 21: Change to "…DOC concentrations have been observed in peatland streams…"

P14 line 13-23: I agree that the effect of hydrology on DOC dynamics is complex and that there is probably a multitude of interactions. One interaction that you do not discuss is the effect of different flow pathways at different discharge conditions (see e.g. Bishop et al., 2004 and Seibert et al., 2009). If you have more organic rich soils close to the soil surface compared to deeper soils, one could expect that concentrations are higher at high stream discharge compared to at low stream discharge. What do the relationship between log(color) and log(discharge) look like? Positive, negative or neither? For Glenamong, which is the only site where you report a similar relationship, this looks complex but generally positive. There are several studies that have looked at such C-Q relationships (see e.g. Creed et al., 2015; Musolff et al., 2015; Moatar et al., 2017; and Winterdahl et al., 2014).

P15 line 23-24: "…warm and dry rather than warm and wet conditions…"

P15 line 26: "…time-series analysis at the annual…"?

P15 line 27: remove the . before "both"

P15 line 30: Colder and drier than what? Change to "...to relatively cold and dry conditions, and dry weather..."

P15 line 31: "Cold conditions..."

P16 line 5: "...minimum annual total DOC yield..."

P16 line 7: New sentence at "However..."

P16 line 8-9: Perhaps true, but your case would be stronger if you could show this with data and statistics – are there any relationships between annual export and e.g. NAO, precipitation or temperature?

P16 line 16-17: Again, this is not something you have shown with data and statistics. However, you may not have the data to actually show this since you only study three streams. I think you should de-emphasize the spatial patterns and concentrate on temporal patterns.

Figure 1: The figure text should start with a capital letter.

Figure 2: There is a parenthesis, which should be removed, at the end of Standardised Precipitation Index on the axis label in A. Also, should the unit for Soil Moisture Deficit (on the left axis) be mm/d?

Figure 4: What are the units on the axes (if any)? Another detail, in previous figures you have indexed sub-figures (panels) with capital letters but now you use lower-case letters. I think you should be consistent.

Figure 5: What is actually displayed in these figures? The text gives some information but there is nothing on the vertical axes – should there be labels and units here? And in e), is that some composite trend (how was that done?) since you write that this is "mean colour concentration in the three sub-catchment rivers"?

Figure 6: I would prefer to use letters to name the different panels instead of writing e.g. "bottom left". There is a ) missing after "top left".

Table 1: The table text should start with a capital letter. Also, what do numbers within parentheses mean? Are those standard dev.? If so, why are you reporting ranges for some parameters but means + std. dev. for others? In addition, I guess the water chemistry data is for stream water but I think it would be good if you clarify this in the caption. What does (312) mean after CORINE Coniferous Forest %?

Table 2: The table text should start with a capital letter. Also, should there be a , after Table 2? Here you write that the data cover 2011-2017 but from the main text I got the impression that data was from 2011-2016. Which is correct? In addition, I think it would be clearer if you used the same acronyms in this table as you use in the main text, i.e. SMD for soil moisture deficit, NAO and only Stemp100 (instead of s(Stemp100)). What does s(...) mean anyway? Is that what is reported by R?

References: Billett, M. F., M. H. Garnett, and F. Harvey (2007), UK peatland streams release old carbon dioxide to the atmosphere and young dissolved organic carbon to rivers, Geophys. Res. Lett., 34, L23401, doi: 10.1029/2007GL031797.

Bishop, K., J. Seibert, S. Köhler, and H. Laudon (2004), Resolving the Double Paradox of rapidly mobilized old water with highly variable responses in runoff chemistry, Hydrological Processes, 18(1), 185-189

Christ, M. J., and M. B. David (1996), Temperature and moisture effects on the production of dissolved organic carbon in a Spodosol, Soil Biology and Biochemistry, 28(9), 1191-1199

Creed, I. F., et al. (2015), The river as a chemostat: fresh perspectives on dissolved organic matter flowing down the river continuum, Canadian Journal of Fisheries and Aquatic Sciences, 72(8), 1272-1285, doi: 10.1139/cjfas-2014-0400.

Evans, C. D., C. Freeman, L. G. Cork, D. N. Thomas, B. Reynolds, M. F. Billett, M. H. Garnett, and D. Norris (2007), Evidence against recent climate-induced destabilisation of soil carbon from 14C analysis of riverine dissolved organic matter, Geophys. Res.

Lett., 34, L07407, doi: 10.1029/2007GL029431.

Ferrari, G. M., M. D. Dowell, S. Grossi, and C. Targa (1996), Relationship between the optical properties of chromophoric dissolved organic matter and total concentration of dissolved organic carbon in the southern Baltic Sea region, Marine Chemistry, 55(3-4), 299-316

Filella, M., and J. Rodríguez-Murillo (2014), Long-term Trends of Organic Carbon Concentrations in Freshwaters: Strengths and Weaknesses of Existing Evidence, Water, 6(5), 1360-1418

Kritzberg, E. S., and S. M. Ekström (2012), Increasing iron concentrations in surface waters - a factor behind brownification?, Biogeosciences, 9(4), 1465-1478, doi: 10.5194/bg-9-1465-2012.

Moatar, F., B. W. Abbott, C. Minaudo, F. Curie, and G. Pinay (2017), Elemental properties, hydrology, and biology interact to shape concentration-discharge curves for carbon, nutrients, sediment, and major ions, Water Resources Research, 53(2), 1270-1287, doi: 10.1002/2016WR019635.

Musolff, A., C. Schmidt, B. Selle, and J. H. Fleckenstein (2015), Catchment controls on solute export, Advances in Water Resources, 86, Part A, 133-146, doi: 10.1016/j.advwatres.2015.09.026.

Neff, J. C., and D. U. Hooper (2002), Vegetation and climate controls on potential CO2, DOC and DON production in northern latitude soils, Global Change Biology, 8(9), 872-884

Seibert, J., T. Grabs, S. Köhler, H. Laudon, M. Winterdahl, and K. Bishop (2009), Linking soil- and stream-water chemistry based on a Riparian Flow-Concentration Integration Model, Hydrology and Earth System Sciences, 13(12), 2287-2297, doi: 10.5194/hess-13-2287-2009.

Temnerud, J., and K. Bishop (2005), Spatial variation of streamwater chemistry in two

Swedish boreal catchments: Implications for environmental assessment, Environmental Science & Technology, 39(6), 1463-1469, doi: 10.1021/es040045q.

Winterdahl, M., M. Erlandsson, M. N. Futter, G. A. Weyhenmeyer, and K. Bishop (2014), Intra-annual variability of organic carbon concentrations in running waters: Drivers along a climatic gradient, Global Biogeochemical Cycles, 28(4), 451-464, doi: 10.1002/2013GB004770.

Winterdahl, M., M. B. Wallin, R. H. Karlsen, H. Laudon, M. Öquist, and S. W. Lyon (2016), Decoupling of carbon dioxide and dissolved organic carbon in boreal headwater streams, Journal of Geophysical Research: Biogeosciences, 121(10), 2630-2651, doi: 10.1002/2016JG003420.

Worrall, F., N. J. K. Howden, T. P. Burt, and R. Bartlett (2018), Declines in the dissolved organic carbon (DOC) concentration and flux from the UK, Journal of Hydrology, 556, 775-789, doi: 10.1016/j.jhydrol.2017.12.001.

---

## Author Comment (AC1) · 16 Dec 2018

Responses to reviewer No 2 comments for Doyle, B. C., de Eyto, E., Dillane, M., Poole, R., McCarthy, V., Ryder, E., and Jennings, E.: Synchrony in catchment stream colour levels is driven by both local and regional climate, Biogeosciences Discuss., https://doi.org/10.5194/bg-2018-272, in review, 2018. Reviewer 2 Comment: Line 5: well, all of this carbon is not transferred to the atmosphere since some of it may be stored in long-term deposits such as lake or ocean sediments. Response: We agree, the sentence now reads. 'Streams draining upland catchments carry large quantities of carbon from terrestrial stocks to downstream freshwater and marine ecosystems. Here it either enters long-term storage in sediments or enters the atmosphere as gaseous carbon through a combination of biotic and abiotic processes.' Comment: Line 10:

Temporal change in what? Response: We agree that the meaning is not clear, the sentence now reads. 'This six-year data set was collected in three contiguous sub-catchments (the Black, the Glenamong and the Srahrevagh) in a blanket peatland catchment system in western Ireland. The data were used to describe the patterns of change in river water colour over time and to assess the main factors explaining these changes.' Comment: Line 12: I guess unit should be mg Pt Co L-1. Response: The units have been corrected in the manuscript. Comment: Line 12-14: I find this sentence a little odd; I expect something to follow the "Although ...". Ok, so the colour concentration was higher in Srahrevagh, but why "although"? Response: We have rewritten, and the sentence now reads. 'At 130 mg Pt Co L-1, the mean colour levels in the Srahrevagh (the subcatchment with lower rainfall and higher forest cover) were almost 50% higher than those from the Black and the Glenamong, which were 95 and 84 mg Pt Co L-1 respectively.' Comment: Line 17-18: Does these numbers (54% and 58%) refer to 1) soil temperature + soil moisture deficit and 2) NAO or to 1) soil temperature and 2) soil moisture and NAO. There are only two numbers but three variables making this sentence unclear. In the next sentence you refer to the combined effect of three variables; why do you not do that here? Response: We agree that sentence was unclear. We have reworded as follows: 'For both the Black and its nested Srahrevagh catchment, three variables (soil temperature, SMD and the weekly NAO) combined to explain 54% and 58% of the deviance in colour respectively'. Comment: Line 21: remove one "each" Response: This was an error, and the sentence now reads: 'Each relationship, however, varied in phase, further highlighting the complexity of the mechanisms driving river colour in the sub-catchments.' Comment: Line 23: You use different number of digits here. Also, I guess these numbers are per km2. So should it be 15.0 and 14.7 t C km-2 yr-1? And why do you only report load for two of the three catchments? Response: We agree. The sentence now reads, 'The estimated mean annual DOC loads for the Black and Glenamong rivers to Lough Feeagh were 15.0 and 14.7 t C km-2 yr-1 respectively, and the export values displayed significant inter-annual variation that was intimately linked to climate variability.' We reported the load

from the Black and the Glenamong only as these are the two main inflows to Lough Feeagh and therefore these are the two main C loadings to the lake. The Srahrevagh sub-catchment is nested within the Black catchment.

Comment: Line 25: but the analysis you refer to above relate to concentrations. At least in the abstract, you do not mention what controls the C export. Response: We agree with the reviewer that the main focus of the study is the variation of colour concentration and therefore DOC concentration in the catchment streams. We report the calculated C export to give readers an idea of the scale of C transport in the sub-catchments. Comment: Introduction P 3 line 5: What do you mean by "...primary production exceeds ... soil organic matter"? Response: We agree that the meaning was not clear. The sentence now reads. 'Under such conditions, primary production exceeds decomposition of soil organic matter, and therefore soil organic carbon (C) accumulates.' Comment: P 3 line 11-12: I suggest you remove the assumed sources of DOC and POC in the parentheses, partly because the sentence is general in form (for instance, not all DOC comes from peat degradation in many catchments) and partly because I am not aware of studies that clearly identify the primary source(s) of DOC or pathway of DOC formation. Previous studies in the UK have e.g. shown that DOC generally is of recent origin, i.e. post 1950s (see e.g. Evans et al., 2007 or Billett et al., 2007). This does not mean that old peat is N sources (could, however, be young peat!). Also P3 line 12: remove "more" before "dominant" Response: We agree to both comments the sentence now reads. 'In most studies which have evaluated fluvial losses of both dissolved organic carbon and particulate organic carbon, DOC has been identified as the dominant C form, representing between 60% and 88% of the total carbon load.' Comment: P 3 line 12-16: You need to be careful with the terminology here. Do you e.g. mean that 60 and 88% of total carbon load, i.e. including DOC, POC, DIC and PIC, were DOC? Or do you mean that 60-88% of total organic C was DOC? Whereas DOC may dominate in many areas, this is often not true where there are large portions of calcareous bedrock within the catchment. Thus, you need to clarify if you mean total C (i.e. including inorganic C) or total organic C. Response: We agree that this

sentence is somewhat confusing the sentence refers to organic C only and that the DOC fraction dominates. The sentence now reads. 'In most studies which have evaluated fluvial losses of both dissolved organic carbon and particulate organic carbon, DOC has been identified as the more dominant C form, representing between 60% and 88% of the total organic carbon load.' Comment: P3 line 20: yes, but not only decomposition but temperature will also affect other potential DOC forming processes, e.g. root exudates from primary producers Response: We agree. The sentence now reads. 'At local scales, temperature affects potential DOC forming processes, including peat decomposition rates and root exudates and therefore the availability of DOC, while higher precipitation increases the washout of DOC from soils.' Comment: P 3 line 26: I guess it is the DOC in the lakes that correlate with climate indices, not the lakes themselves, or? Response: We agree the meaning was not clear. The sentence has been corrected and it now reads. 'At regional scales, DOC concentrations have been shown to be influenced by global weather patterns, for example DOC concentrations in certain Canadian lakes were shown to correlate with climate indices such as the Pacific Decadal Oscillation and the Southern Oscillation Index (Zhang et al., 2010).' Comment: P3 line 29: remove the acronym SLP – it is not being used anywhere else in the manuscript and thus superfluous. Response: We agree and it has been removed from manuscript. Comment: P 3 line 32: What should Ref be? Response: It was a typo and it has been removed from manuscript Comment: P4 line 14-16: This was also found by Winterdahl et al. (2014) where TOC was increasing in about half of 130 streams, but without any clear geographical patterns. Also worth mentioning is that some authors claim that DOC concentrations have stopped increasing or are actually decreasing (Worrall et al., 2017) whereas others have pointed to methodological differences among studies that limit interpretations of potential trends (Filella & Rodriguez-Murillo, 2014). And P4 line 16: But the studies referred above did not study DOC export but DOC concentrations. Once again you need to make clear if it is export or concentrations that are the focus of study. The export is to a large extent controlled by water discharge, and thus ultimately by the difference between precipitation and evapotranspiration. Response: We agree and propose to rewrite as follows. 'There are, however, also studies where DOC concentrations have been shown to have decreased (Clair et al., 2008; Worrall et al., 2017), or no increase has been observed, such as within certain catchments in the U.K. (Worrall and Burt, 2007). Winterdahl et al. (2014) also reported increases in TOC in only half of 130 Swedish streams, but with no clear geographic pattern highlighting the need for further examination of the complex relationship between DOC concentration and climate.' Comment: P4 line 21-22: This sentence seems out of topic – you have not discussed effects of changes in nutrient cycles before and Kurbatova et al. studied Russian bogs which I suspect behave quite differently compared to the blanket peats on the British Islands (in terms of e.g. hydrology and topography). Response: We agree and the sentence has been removed from the manuscript. Comment: P4 line 28-29: Repetition. You have already mentioned that this is one of your study catchments.

Response: We agree, the portion of the sentence has been removed. The sentence now reads. 'For the Glenamong sub-catchment, Ryder et al. (2014) previously reported that soil temperature, river discharge and a dry spring period explained approximately 60% of the deviance in DOC concentrations over a two-year period.' Comment: P4 line 32: change to "...climatic conditions, e.g. the NAO, as a possible..." Response: We agree, and the sentence now reads. 'The present study expands on that work, firstly by comparing colour concentrations from three contiguous peat sub-catchments that differ in their catchment characteristics, and secondly by including the role of the regional climatic conditions, e.g. the NAO, as a possible driver.' Comment: P4-P5 line 33 ff: The aims need clarification. First you mention the Burrishoole catchment but later you write "water colour from rivers in three sub-catchments in a blanket peatland catchment" – why not specify that this is the Burrishoole catchment? Also, part 2 need to be specified; the effects of main climatic drivers on what? Response: We have clarified the aims of the paper. The sentence now reads: 'The principal aims of the current study, using water colour data from the Burrishoole catchment in the west of Ireland were 1, to compare the sub-seasonal, seasonal and multi-annual trends

in water colour 2, to identify the main climatic drivers of water colour variation and 3, to quantify the inter-annual variability in fluvial export of DOC over the study period.' Comment: P5 line 6-8: It would be interesting to know the area of the entire Burrishoole catchment. Response: This information has been added, the sentence now reads. 'The Burrishoole catchment (∼100 km2) is a topographic basin, that has been carved into the Nephin Beg mountain range over successive ice-ages and comprises twenty-one lakes of sizes ranging from 0.04 ha to 395 ha and approximately 143 kilometres of interconnecting rivers and streams (53° 55' N 9° 55' W).' Comment: P5 line 18: You can remove the (Co-ORdinated INformation on the Environment) but write CORINE in capital letters (as you do in the reference list). Response: We agree. The sentence now reads. 'Land cover in the catchment comprises 52% blanket peat, 15% forestry, with the remaining 33% being made up of discrete parcels of transitional woodland and scrub, natural grasslands and agricultural land (CORINE, 2012)' Comment: P5 line 24: why do you report precipitation only for 2010-2016 when you obviously have a longer time series of precipitation from the area? The mean precipitation for 2010-2016 is reported in the results anyway. Response: We agree. We are adding a recently published long term averages from the Newport Meteorological Station. The sentence now reads. 'Long-term average annual precipitation at this station (1960–2014) was 1564 mm. Average daily rainfall for the same period was 4.3 mm (±6.2 mm SD), and 75% of days had some measurable rainfall (de Eyto et al., 2016)' Comment: P5 line 26: you repeat "spatially" here – remove one P5 line 27: Above you did not use a thousand separator (,) but here you do. You need to be consistent and comply with the format of the journal. Response: We agree, one "spatially" was removed and the thousand separator has been removed also. Comment:

P5 lines 26-28: Are these precipitation numbers from the same year? Or are they annual means? That is not clear now. I think you need to show the spatial variability better because as it is now, it is not clear how these observations differ from Newport (besides that the numbers are a bit different). You could perhaps show how large the spatial differences are on average among years (including all three stations with precip.

Response: They are annual means over the study period. We have included standard deviations for the rain gauge data. The sentence has been reworded as follows. 'It is also important to note that spatially, rainfall levels varied spatially across the catchment over the study years from an annual average of 2623 mm year-1 ($\pm$386 mm year-1 SD), recorded at an automatic rain gauge in the northwest of the catchment (Namaroon) to 1508 mm year-1 ($\pm$158 mm year-1 SD) at the south of the catchment (Millrace rain guage) (MI unpublished data).' Comment: P6 line 8: I guess this should be (Figure 1 and Table 1) Response: We have amended the manuscript. Comment: P6 line 16: change to "three sub-catchments" Response: We have amended the manuscript. Comment: P6 line 23: Change to "10 m resolution" Response: We have amended the manuscript. Comment: P6 lines 26-27: What was the precision of this instrument? If you have data on accuracy, that would also be relevant to report here. Response: We have included the accuracy of the instrument; 'Colour (mg PtCO L-1) was measured within hours of sampling using a HACH Dr 2000 spectrophotometer at 455 nm on water filtered through Whatman GF/C filters (pore size: 1.22 $\mu$m). Wavelength accuracy = $\pm$2 nm from 400 – 700 nm and $\pm$3nm from 700 – 900 nm.' Comment: P6 line 29: "Daily precipitation and soil temperature data. . ." Response: We have amended the manuscript. Comment: P6 line 30- ff: How was the rating curve calibrated? I.e. what methods were used to construct the rating curve? I also think you should report the accuracy of this rating curve, e.g. with an R2. Response: We have added this information to the manuscript, the sentence reads as follows: 'The levels for the Glenamong and Srahrevagh rivers were converted to volume of discharge per second (m3 s-1) using site specific ratings curves that have been developed using stream-flow data collected regularly, Glenamong R2 = 0.98, Srahrevagh R2 = 0.96 (Marine Institute unpublished data).' Comment: P7 line 6-8: Why two different tests? Response: We have reworded the sentence to clarify the reason for two different statistical tests, the sentence now reads. 'As the Black and Srahrevagh rivers are in the same river system, a non-parametric Wilcoxon Signed Rank Test was used to test for statistical differences between their colour concentrations.' Comment: P7 line 16-17: This sentence needs to be rephrased. Should the second "for" be removed? Response: We agree, the sentence was poorly phrased, it now reads. 'For a specific time series, colour concentration in surface waters can be expressed as:' Comment: P7 line 21: I know many authors equal colored DOM and DOC but since only a (small) fraction of DOM actually is colored (see e.g. Ferrari et al., 1996) you might want to refer to CDOM here (and at other places where you use color to draw conclusions about DOC) instead. Response: We have found excellent relationships between DOC and colour in the Glenamong river (e.g. r2 = 0.88, p $\leq$ 0.001, n = 366) and more recently in 2017 (r2 = 0.95, p $\leq$ 0.001, n = 13) Comment: P7 line 23: General additive mixed models – this section is a little dense. Could you please divide it into a few paragraphs? Response: We agree. The section has been divided into 3 paragraphs in the manuscript: Comment: P8 line 6: Unit should be (m s-1). Response: We agree and the amendment has been made to manuscript Comment: P8 line 7: So the humidity here should actually be "relative humidity" Response: We agree and the amendment made to manuscript Comment: P8 line 8: How was actual evaporation estimated? Response: Soil moisture deficit (SMD) was calculated using a procedure described by Brereton et al. (1996) for Irish grasslands. Potential evapotranspiration rates were estimated based on air temperature and sunshine data using the method of Priestley and Taylor (1972) and then actual evapotranspitaion was calculated as a proportion of potential evapotranspiration based on Aslyng (1965) Comment: P8 line 18: I guess SMD is soil moisture deficit but the acronym has not been defined. Response: We agree. Soil Moisture Deficit (SMD) is now defined where it is introduced in the manuscript. Comment: P8 line 30: This should be rephrased. Water color was not converted but DOC was estimated from water color. I think you need to be clearer about this throughout the manuscript. Response: We agree. The sentence has been rephrased and now read. 'DOC (mg L-1) concentration was estimated from water colour concentration (PtCo mg L-1) using a linear model developed between water colour and DOC from the Glenamong River between April 2010 and September 2011.' Comment: P9 line 1: There should be a . after "sub-catchment rivers". Response: We agree, the full stop has been added. Comment:

[Figure]

P9 line 2: Is this really the accuracy? Or is it the precision? If this is the accuracy, what then is the precision? Also, on this line it should be "5 ppb". Response: The precision of the instrument has been added: 'DOC analysis was carried out using a Sievers 5310C Total Organic Carbon analyser (Range 4 ppb to 50 ppm, accuracy $\pm$ 2% or $\pm$ 5 ppb, whichever is greater; precision < 1% relative standard deviation).' Comment: P9 line 3: New paragraph before "Mean annual yield. . ." Response: We agree, the amendment has been made to manuscript. Comment: P9 line 4-5: This sentence should be moved so that it precedes the previous sentence, i.e. first this sentence (starting with "The mean annual load. . .") and then the sentence starting with "Mean annual yield. . ." Response: We agreed, the sentences have been moved and the section reads as follows: 'The mean annual load was calculated by multiplying the calculated stream discharge volume for each week by the weekly DOC concentration and summing the totals. Mean annual yield (per km2) was estimated by dividing the mean annual load by the upstream drainage-basin area.' Comment: P9 line 5: I am a bit confused here, but I think you mean the estimated DOC here, right? Response: We agree, the sentence was unclear. It now reads as follows. 'The mean annual loads were calculated for the Black and Glenamong sub-catchments by multiplying the calculated stream discharge volume for each week by the weekly estimated DOC concentration and summing the totals.' Comment: P9 line 8: Should it be ". . .year, with 2013 being the driest year, with. . ." or something similar? Response: We agree, the sentence now reads as follows. 'Weather conditions varied during the six study years. 2013 was the driest year, with a mean daily precipitation of 3.7 mm day-1 and an annual total precipitation of 1315 mm year-1.' Comment: P9 line 16-17: Do you really have the precision to report these numbers with one decimal? Above you did not use a decimal and I think you should be consistent here (also, how many decimals are realistic based on your measurement equipment?). Response: We agree with the reviewer, decimals have been removed and the sentence now reads: 'The driest summer over the study period occurred in 2013 with 259 mm accumulated rainfall. The driest winter was also in 2012/2013 with 430 mm accumulated rainfall.' Comment: P9 line 28: This would be easier to see if you

also report the specific discharge in e.g. mm/d.

Response: We consider that the discharge units of m3 s-1 is more intuitive for the readers. Comment: P10 line 5: But the cumulative SMD should have unit mm (only), right? Response: We agree, the sentence has been amended and now reads. 'The year with the greatest cumulative SMD was 2013 with an average daily deficit of 8.3 mm. The cumulative SMD reached a maximum of 66.2 mm in July.' Comment: P10 line 20-22: This "random component" does not seem to be entirely random, at least not from what I can tell from figure 3D. How does the autocorrelation of this random component look like? Would it be possible to subtract even more information from this time series (though I have no idea how to do that)? Response: We agree with the reviewer that the decomposed trend does not appear to be entirely 'random'. The random component is so named because it is the component remaining after the seasonal and multi-annual trends have been subtracted. This component appears to correspond with flood and drought ('random' in time) events over the study period. It is our understanding that these decomposed trends are strongly autocorrelated and can really only be compared visually, i.e. no further statistical analysis can be carried out on them. Comment: P10 line 25-28: This information seems misplaced. Why not combine this with the text in the beginning of this section where you also refer to which stream having the highest concentrations? Response: We agree with the reviewer. This section has been moved to the beginning of the section in the manuscript. Comment: P11 line 13: ". . .the optimal model. . ."?

Response: We agree, the sentence has been rephrased and now reads. 'The optimal GAMM for colour in the Glenamong River also had three smoothers, but differed in that it included the log of river discharge rather than SMD' Comment: P11 line 31: Not sure if I agree about discharge here. Based on figure 5, NAO, soil temp and water color seem similar but the increasing trend in discharge starts more than a year after the increase in NAO. Response: We agree with the reviewer that the 'dip' in the trend in water discharge starts more than a year after the NAO. Our aim here is to emphasise

how the trends in NAO, temperature, discharge and water colour have broadly similar patterns, however the timing was different for each. We propose the following change to the sentence. 'The multi-annual trend plots for the NAO, soil temperature, discharge and water colour all had broadly similar patterns that included a distinct dip in the period from late 2012 to late 2013 and a general upward trend after these low points (Fig. 5).' Comment: P12 line 20-25: This section is unclear. Is the second set of numbers reported (i.e. 18.5 and 11.8 t C km2 yr-1) averages among all the study streams? It is not clear how these differ from the first set of numbers (which apparently were for individual streams in individual years). Everything becomes clear when looking at Table 3 but it should be clear from the text as well. Also, sometimes you use the term yield and sometimes load – do these mean different things here? & P12 line 24: This sentence is a bit confusing. I think you should change this to ". . .while 2013 had the least total DOC load. . .". You have already reported that 2013 was the driest year – there is no need to reiterate that here. Response: We agree with the reviewer that the section is somewhat unclear. We propose to reword this section as follows. 'There was a wide range in the annual estimated loads exported from the three sub-catchments over the six study years. These ranged from a maximum DOC load of 38.6 t C km2 year -1 for the Srahrevagh sub-catchment during 2015, almost four times the minimum load of 11.6 t C km2 year -1 exported from the Glenamong sub-catchment in 2013. Also notable was the inter-annual variability of the total calculated load from all sub-catchments, whereby 2011 had the greatest total DOC load of 18.5 t C km2 year -1 while 2013 has the least total DOC load of 11.8 t C km2 year -1. Comment: P12 line 28-29: You should rephrase this sentence. It is unclear, probably because of the misplaced modifier "which" that refer to "DOC levels" or possible to "water colour" in this case. I also think you should change the statement "explained circa 60%" to "explained between 54 and 66 %". Response: We agree with the reviewer and we have amended the sentence as follows. 'This study highlighted the dominant influence of local and regional climate on water colour, which as a proxy for DOC levels, explained between 54 and 66 % of the variability in all three datasets, and the strong synchronicity in these climate signals across the Burrishoole catchment.' Comment: P13 line 2-3: Though I suspect you are right, do you have data to confirm this statement? Do you e.g. have fluorescence data that indicate that DOC primarily is of terrestrial origin? If not, I think you should be more careful and write something like ". . .probably originates primarily from the surrounding catchment. . .". Response: We agree with the reviewer, and we have amended the sentence as follows. 'Colour, and therefore DOC, in these headwater rivers probably originates almost exclusively from the surrounding catchment soils' Comment: P13 line 3-6: If I understand this sentence correctly you claim that you have shown that the DOC export from the different study catchments in your study are related to catchment properties, land use, runoff etc. But this is incorrect, you have not shown this. There are no data that show these relationships. Response: We agree with the reviewer and we have rephrased this sentence, removing the reference to DOC export. The sentence now reads as follows. 'Colour, and therefore DOC, in these headwater rivers probably originates almost exclusively from the surrounding catchment soils and the consistent difference in colour concentrations between each sub-catchment during the study was most likely a function of individual sub-catchment properties such as the extent of peat within catchments (Hope et al., 1997a), land use (Findlay et al, 2001), local runoff (Dillon and Molot., 2005) vegetation type (Sobek et al., 2007) and the unique morphology and geology of the sub-catchment landscape (Moore, 1998).' Comment: P13 line 8-10: But you do not present such an analysis – you only have three sites, so the statistics will be a little shaky, but can you find any of these relationships that you mention? Figures in an appendix could be enough to show if there are any relationship between water color and e.g. the extent of peat soils in the catchment. Response: We agree with the reviewer and we have rephrased the sentence to remove any mention of spatial analysis, the new wording is a follows. 'The extent of peat soils in the study catchments, the length of streams intersecting the peat, slope analysis and CORINE land cover in each sub-catchment (Table 1) may help in explaining the higher levels of colour found in the Srahrevagh'. Comment: P13 line 14-17: Perhaps, but other studies have not found any clear downstream patterns

in DOC concentration (see e.g. Temnerud & Bishop, 2005 and Creed et al., 2015) or clear signs of DOC degradation as water moves downstream in a stream network (see e.g. Winterdahl et al., 2016). Response: We agree with the reviewer that there are conflicting bodies of work on this point. We propose to add the following sentence to the end of the section: 'An additional factor that may have influenced the variation in colour between the sub-catchments could be the distance between a given sampling point and the source of any coloured compounds. Dawson et al. (2002) observed decreases in TOC (both DOC and POC) concentrations in the Upper Hafren (a head-water stream in mid-Wales) downstream from the source that were stated to be related to a decrease in peat depth with altitude, combined with in-stream processing of DOC. A similar process may contribute to the difference in concentration between the up-stream Srahrevagh and downstream Black sampling points. There are however other studies that suggest no clear change in DOC concentration or degradation as water travels downstream (Temnerud & Bishop, 2005 and Creed et al., 2015, Winterdahl et al., 2016).'

Comment: P13 line 21-23: Is it necessary to reiterate the results here? & P13 line 24: Wouldn't Christ and David (1996) and Neff and Hooper (2002) be more relevant references here since they have actually looked at the relationship between temper-ature and DOC "production/leaching". Response: We agree with the reviewer. We have amended the sentence and added the suggested references. It now reads as follows. 'Soil temperature was common to all three GAMMs, and was the dominant explanatory variable, emphasising how dissolved organic carbon is released by peat soils via decomposition processes that are temperature dependant (Christ and David, 1996; Neff and Hooper, 2002).' Comment: P13 line 28-30: You touch upon this but it could perhaps be clarified. You need to think of what you, and most other scientists in this business, refer to as "DOC production" as two different processes (if we simplify everything and ignore e.g. sorption dynamics, solution/dissolution due to changes in water chemistry etc.): 1) the actual DOC production, i.e. some process that forms DOC (could be e.g. exudation of organic molecules through roots or microbial degradation

of solid organic matter), and 2) transport of DOC along active flow pathways in the soil. Process 1) could be active as long as there is water in the soil, even if this water is not moving. Process 2) only happens when the water is actually moving. That is, you could have an area with stagnant soil water where DOC production (process 1) forms a "stock" of DOC that is transported to a nearby surface water body as soon as the flow pathways are activated. Response: We agree with the reviewer on this point. We propose to rewrite as follows: 'The lowered water table, however, reduces the hydrological connection, i.e. the transport of DOC along active flow pathways in the soil (Ryder et al., 2014) This breaks the connection between the source of DOC production and its eventual destination.' Comment: P13-14 line 33-1: But this is not generally the case for DOC, see e.g. data from about 130 streams in Winterdahl et al. (2014) where there is no relationship between seasonality and DOC concentration.

Response: We agree with the reviewer and we propose to rewrite as follows. The strong relationship found between soil temperature and water colour concentrations in the three rivers, and the significant and high common power with river colour at the yearly time scale in the cross-wavelet analysis, indicated that soil temperature was the primary driver of the seasonal pattern in water colour during the study period. It is interesting to note that in general no relationship between seasonality and DOC concentration has been reported from some other studies commonly observed (e.g. Winterdahl et al. 2014). However, our results are consistent with observations of DOC dynamics in some surface waters in temperate peatlands, where seasonal variation has been found to be the largest source of DOC variation in catchments with high DOC concentrations (Clark et al. 2010; Ryder et al., 2014). Comment: P14 line 6: As I understand this, you mean that since concentrations decrease, the export will also decrease. In this case, this is probably true since if soil moisture decreases, stream discharge will also likely decrease. But generally, you can have decreasing concentrations but increasing export if discharge increases. Since discharge on event scales can vary by several orders of magnitude whereas concentrations seldom vary by more than a factor 10, discharge often control the export dynamics, at least on short

time scales. Therefore, I think you should remove ". . .and therefore export. . ." here. Response: We agree with the reviewer, we have amended the sentence and it now reads: reads. 'The relationship of colour with SMD in the Black and Srahrevagh optimum GAMM models indicated that as soil moisture decreased DOC concentrations also decreased.' Comment: P14 line 21: Change to ". . .DOC concentrations have been observed in peatland streams. . ." Response: We agree with the reviewer, the sentence has been amended and it now reads. 'However, immediately following periods of dry weather or drought, pronounced increases in DOC concentrations have been observed in peatland streams'

Comment: P14 line 13-23: I agree that the effect of hydrology on DOC dynamics is complex and that there is probably a multitude of interactions. One interaction that you do not discuss is the effect of different flow pathways at different discharge conditions (see e.g. Bishop et al., 2004 and Seibert et al., 2009). If you have more organic rich soils close to the soil surface compared to deeper soils, one could expect that concentrations are higher at high stream discharge compared to at low stream discharge. What do the relationship between log(color) and log(discharge) look like? Positive, negative or neither? For Glenamong, which is the only site where you report a similar relationship, this looks complex but generally positive. There are several studies that have looked at such C-Q relationships (see e.g. Creed et al., 2015; Musolff et al., 2015; Moatar et al., 2017; and Winterdahl et al., 2014). Response: We agree that this is indeed an interesting interaction, however as discharge was found to be in the optimal model for the Glenamong sub-catchment only, we do not intend to discuss different flow pathways at different discharge conditions further in the manuscript. Comment: P15 line 23-24: ". . .warm and dry rather than warm and wet conditions. . ." Response: We agree with the reviewer and the sentence has been amended, it now reads. 'However, some studies have also suggested that positive phases of the NAO during the summer are associated with warm and dry rather than warm and wet conditions over northwest Europe in particular the UK and much of Scandinavia (Folland et al. 2008).' Comment: P15 line 26: ". . .time-series analysis at the annual. . ."? & P15 line 27:

remove the . before "both" Response: We agree with the reviewer and the sentence has been amended, it now reads. 'However, the negative relationship apparent in the cross-wavelet time-series analysis at the annual time step may also merely reflect the fact that both time series have seasonal patterns, but are not linked by any causal mechanism.'

Comment: P15 line 30: Colder and drier than what? Change to ". . .to relatively cold and dry conditions, and dry weather. . ." Response: We agree with the reviewer and the sentence has been amended, it now reads. 'Negative NAO values during the winter generally correspond to relatively cold and dry conditions, and dry weather was observed throughout 2013, reflected in the SPI Index, beginning during the winter of 2012/2013.' Comment: P15 line 31: "Cold conditions. . ." Response: We agree with the reviewer and the sentence now reads. 'Cold conditions were also confirmed by the sharp dip in the multi-annual trend of soil temperature observed during the same winter period.' Comment: P16 line 5: ". . .minimum annual total DOC yield. . ." Response: We agree with the reviewer and the sentence now reads. The minimum annual total DOC yield from the Burrishoole catchment was. . . Comment: P16 line 7: New sentence at "However. . ." Response: We agree and have amended that manuscript. Comment: P16 line 8-9: Perhaps true, but your case would be stronger if you could show this with data and statistics – are there any relationships between annual export and e.g. NAO, precipitation or temperature? Response: We agree with the reviewer and have amended that manuscript to remove any references to carbon export being linked to climate factors.

Comment: P16 line 16-17: Again, this is not something you have shown with data and statistics. However, you may not have the data to actually show this since you only study three streams. I think you should de-emphasize the spatial patterns and concentrate on temporal patterns. Response: We agree with the reviewer and have reworded sections of the discussion to concentrate on temporal patterns only, and de-emphasising spatial patterns. 'The results of this study emphasised how colour

concentrations, and therefore DOC levels, respond to common climatic drivers which operate at both a local and regional scale.' Figures and Tables: Comment: Figure 1: The figure text should start with a capital letter. Response: This has been amended in the Figure. Comment: Figure 2: There is a parenthesis, which should be removed, at the end of Standardised Precipitation Index on the axis label in A. Also, should the unit for Soil Moisture Deficit (on the left axis) be mm/d? Response: Parenthesis has been removed and the units on the left axis have been corrected. Comment: Figure 4: What are the units on the axes (if any)? Another detail, in previous figures you have indexed sub-figures (panels) with capital letters but now you use lower-case letters. I think you should be consistent. Response: Units have been added to the figure and the panels have been indexed with upper-case letters. Comment: Figure 5: What is actually displayed in these figures? The text gives some information but there is nothing on the vertical axes – should there be labels and units here? And in e), is that some composite trend (how was that done?) since you write that this is "mean colour concentration in the three sub-catchment rivers"? Response: Labels and units have been added on the vertical axes for each panel. The bottom panel (E) shows the decomposed multi-annual trend (STL) of mean colour concentrations from all three catchments over the study period. Comment: Figure 6: I would prefer to use letters to name the different panels instead of writing e.g. "bottom left". There is a ) missing after "top left". Response: We have added index letters A, B, C and D to Figure 6 and updated the caption to correspond with the figure. Comment: Table 1: The table text should start with a capital letter. Also, what do numbers within parentheses mean? Are those standard dev.? If so, why are you reporting ranges for some parameters but means + std. dev. for others? In addition, I guess the water chemistry data is for stream water but I think it would be good if you clarify this in the caption. What does (312) mean after CORINE Coniferous Forest %? Response: Table text has been revised to start with a capital. The numbers in parentheses are standard deviations, the caption has been updated to reflect this. Stream water chemistry has been added to the caption to clarify. The (312) has been removed from the table. Comment: Table 2: The table text should start with a

capital letter. Also, should there be a , after Table 2? Here you write that the data cover 2011-2017 but from the main text I got the impression that data was from 2011-2016. Which is correct? In addition, I think it would be clearer if you used the same acronyms in this table as you use in the main text, i.e. SMD for soil moisture deficit, NAO and only Stemp100 (instead of s(Stemp100)). What does s(. . .) mean anyway? Is that what is reported by R? Response: Table text has been revised to start with a capital. Comma has been added. Date range has been revised in the table (2011 – 2016). We agree with the reviewer and the acronyms in the table have been amended to correspond with the main text. Comment: Typos p. 2, L. 21: delete one of the "each" p. 4, L. 10: "trend" should be plural to be consistent with "changes" mentioned before p. 5, L. 26: delete one of the "spatially" p. 6, L. 29 replace the first "," by "and", and remove the second "," p. 7, L. 16: delete one of the "for" p. 8, L. 23: use "were" instead of "are" p. 9, L. 1: add a full stop between "rivers" and "doc" p. 15 , L. 27: remove the full stop between "that" and "both" Fig. 2: remove "(" after "standardized precipitation index" at the y-axis label of the uppermost panel. p. 24, L. 4: there is a digit missing in "201" Table 1 caption: "sub-catchmen" is missing a "t" Response: All the above typos have been corrected. References: Aslyng H. C., 1965. Evaporation, evapotranspiration and water balance investigations at Copenhagen 1955–64. Acta Agric. Scand., 15: 284-300. Brereton, A .J., S. A. Danilov and D. Scott, 1996. Agrometerology of grass and grasslands in middle latitudes. Technical note no. 197. World Meterological Organization, Geneva, p. 36. Priestley C. H. B. and R. J. Taylor, 1972, On the assessment of surface heat flux and evaporation using large-scale parameters. Mon. Weather Rev.100: 81-92. Ryder, E., de Eyto, E., Dillane, M., Poole, R., and Jennings, E.: Identifying the role of environmental drivers in organic carbon export from a forested peat catchment, Sci. Total Environ., 490: 28–36, 2014.

---

## Author Comment (AC2) · 16 Dec 2018

Responses to reviewer No 1' Comment:s for Doyle, B. C., de Eyto, E., Dillane, M., Poole, R., McCarthy, V., Ryder, E., and Jennings, E.: Synchrony in catchment stream colour levels is driven by both local and regional climate, Biogeosciences Discuss., https://doi.org/10.5194/bg-2018-272, in review, 2018.

Reviewer 1

Comment:

Specific Comment:s to the authors - Throughout the manuscript: The authors often use terms such as "controls" and "drivers" (see e.g. p. 2, L. 11). These terms imply mechanistic relationships between environmental drivers and water color. However, the

authors used a statistical approach that allows to investigate correlations, not mechanistic links. I suggest to rephrase all terms throughout the manuscript to make clear that relationships were correlative, not mechanistic.

Response:

We agree generally with the reviewer on this point. We have gone through the manuscript and replaced with terms that reflect the statistical nature of our analysis in many places. These include, for example, p2 line 11: 'and used to assess the effect of individual catchment characteristics and identify the drivers that best explained observed temporal change in river colour.' Another example at p2 line 13: 'General additive mixed modelling was used to identify the principle environmental drivers that explained a significant percentage of the deviance in colour levels in the rivers.' However, we would also argue that there are well recognised mechanistic links between local weather and the concentration of humic material in surface waters in peat catchments (Clark et al., 2008; Ryder et al., 2014; Ritson et al., 2017). Therefore where required and particularly in relation to this analysis we use the term drivers which we now have defined in the methods section as follows: p. 7 L24: 'To identify the main explanatory factors, which we refer to as drivers, of colour, general additive mixed models (GAMM) with cubic smoothing regression splines and Gaussian distributions were developed using the mgcv package (Wood, 2006).'

Comment:

p. 2, L. 1 (Title): Here, the term "climate" is used, but in the abstract (L. 25) the term "meteorological drivers". Please harmonize.

Response:

To address this point, the text now reads. 'The results of the study highlighted the interaction of catchment characteristics and local and regional climate in controlling aquatic carbon export. The important role of temperature, and past and current precipitation, in

particular, show the vulnerability of blanket peatland carbon stores to projected climate change.'

Comment::

p. 2, L. 4: the term "reservoirs" could be misunderstood, especially by the aquatic biogeochemistry community. Maybe simply use the term "stocks", or "soils" instead of "terrestrial reservoirs"?

Response:

We agree. The text now reads. 'significant loads of carbon from terrestrial stocks to downstream freshwater and marine aquatic ecosystems'

Comment:

p. 2, L. 7-10: This is a very long sentence and hard to digest. I suggest to split it.

Response:

We agree. The sentence now reads. 'We analysed sub-annual and inter-annual changes in river water colour (a reliable proxy measurement of dissolved organic carbon (DOC)) using six years of weekly data, from 2011 to 2016. This time-series data set was gathered from three contiguous river sub-catchments, the Black, the Glenamong and the Srahrevagh, in a blanket peatland catchment system.'

Comment:

P. 2, L. 12: maybe clarify more by adding "in correlations" after "frequencies"?

Response:

We agree – the sentence now reads. 'while wavelet cross correlation analysis was used to identify common frequencies in correlations.'

Comment:

P. 2, L. 12-14: "Although at 130 mg PtCo L-1, the colour levels in the Srahrevagh (the subcatchment with lower rainfall and higher forest cover) were almost 50% higher than those from the Black and Glenamong, 95 and 84 mg Pt Co L-1 respectively." Why do the authors introduce the sentence with "although"? is it to highlight that the low rainfall catchment was expected to have clearer water than the other catchments? I would restructure the sentence to get this message better come through.

Response:

Agreed - This sentence now reads: 'At 130 mg Pt Co L-1, the colour levels in the Srahrevagh (the subcatchment with lower rainfall and higher forest cover) were almost 50% higher than those from the Black and Glenamong, which were 95 and 84 mg Pt Co L-1 respectively.'

Comment:

p. 2, L. 15-16: "illustrating that environmental drivers operated synchronously at each of these temporal scales, and also spatially within the same catchment ": what exactly do the authors want to state here? It reads to me like that environmental drivers were similar across the catchments, but this would contrast to the conclusion that drivers varied depending on catchment-specific characteristics. It would also contrast the statement further down in the abstract (L. 24-25) that "the results of the study highlight the interaction of catchment conditions and regional meteorological drivers". Please clarify.

Response:

We agree with the reviewer that the differences in spatial characteristics for a small sample of three sub-catchments was not sufficient to undertake statistical analysis and therefore to make robust conclusions. We have removed references to spatial analysis between the sub-catchments from the manuscript due to lack of statistical support, with the exception of noting the statistical difference in colour levels in the Srahrevagh (see point immediately above). We propose that the sentence referred

to will now read: '…illustrating that environmental drivers operated synchronously at each of these temporal scales'

Comment:

p. 2, L. 23: why is the term "although" used here? Isn't it enough to simply write that there was inter-annual variation?

And

p. 2, L. 24: it would be interesting for a wide readership to know whether these interannual variations in DOC loads are linked to variability in climate. This remains unclear in the way it is phrased here.

Response:

We agree with the two points above. The word 'although' has been removed. The inter-annual variation in DOC loads are undoubtedly linked to climate, and more specifically variation in rainfall however we have not statistically tested for this. This sentence now reads: 'The estimated mean annual DOC loads exported from the Black and Glenamong rivers to Lough Feeagh were 15 t C km2 yr-1 and 14.7 t C km2 yr-1 respectively. The annual export values over the six years displayed significant inter-annual variation that was most likely linked to climate variability.'

Comment:

p. 2, L. 24: Can the authors specify what is meant with "interaction of catchment conditions and regional meteorological drivers? what characteristics makes DOC export from a catchment more or less susceptible to environmental drivers? This should be highlighted here, or at least in the conclusions of the manuscript, if supported by the data.

Response:

We agree that this sentence is unclear. As noted above, we have removed references

to any spatial analysis between sub-catchments and any conclusions related to that. This sentence now reads: 'The results of the study highlighted the role of climate in controlling stream water DOC concentrations, and aquatic carbon export, and therefore the vulnerability of blanket peatland carbon stores to future changes in temperature and precipitation.'

Comment:

p. 3, L. 3: "warmer and wetter" conditions is relative. Which climate zone is referred to here?

Response:

We agree. The sentence now reads: 'Blanket peat ecosystems occur within a narrow window of climatic conditions, characterised by relatively warmer and wetter conditions, in temperate regions where precipitation exceeds potential evaporation by a ratio of about three to one.'

Comment:

p. 3, L. 19: what are "year-to-year changes in climate"? climate refers to a period of at least 30 years. I think it is meteorological conditions the authors refer to here.

Response:

We agree. The sentence now reads: 'Longer term patterns in DOC concentrations or in proxies for DOC have been linked to year-to-year changes in meteorological conditions at both local and regional scales.'

Comment:

p. 3. L. 19-25: is an introduction of these enzymatic mechanisms needed? The terms used are quite technical and it seems that it is not relevant for the remainder of the manuscript

Response:

Agreed. The sentence has been removed from the manuscript.

Comment:

p. 3, L. 26: "Canadian lakes have been shown to correlate": what property of these lakes is referred to here? Response:

We agree that the meaning was not clear. The sentence now reads: 'At regional scales, DOC concentrations have been shown to be influenced by global weather patterns, for example DOC concentrations in certain Canadian lakes have been shown to correlate with climate indices such as the Pacific Decadal Oscillation and the Southern Oscillation Index (Zhang et al., 2010)'.

Comment:

p. 3, L. 32: which "Ref" is referred to here?

Response:

Apologies. This was a typo and has been removed from the manuscript.

Comment:

p. 4, L. 25: the authors mention here the implications for future management of peatland systems. Can the authors formulate such implications in the discussion section?

Response:

As we consider that peatland management is is outside the scope of this manuscript, we think it best to remove this part of the sentence entirely. The sentence now reads: 'Examining riverine fluxes of carbon from these catchments provided a means to quantify the export of C from long-term storage in peatland ecosystems and to explore the effects of climatic variables on these C stores'.

Comment:

p. 6, L. 11: define "blanket peat" (regarding peat depth) when first mentioned in the manuscript

Response:.

The definition has been moved to P3 L. 6&7

Comment:

p. 6, L. 18+20: "gentle" and "steeper" slopes are relative terms. I suggest to refer to absolute numbers here.

Response:

This sentence now reads. 'The Srahrevagh sub-catchment has the greatest proportion of slopes with gradients ranging between 0 and 20%, while the Glenamong is the most mountainous of the three sub-catchments, having the greatest altitude range and containing the greatest proportion of slopes steeper than 50% (Table 1).'

Comment:

p. 6, L. 23: Please add a reference or vendor for the Arcmap program.

Response:

The vendor has been added: 'ArcMap 10.3.1. Environmental Systems Research Institute (ESRI)'

Comment:

p. 6, L. 29: Is it the Newport Met Station that is indicated in Fig. 1? If so, please indicate in the figure and refer to the figure in the text.

Response:

Yes, Newport Met Station is in Figure 1, the figure has been amended.

Comment:

p. 6, L. 30: please give the location of the vendor of the water level loggers.

Response:

This is added as 'OTT Hydrometry Orpheus Mini water level loggers'.

Comment:

p. 6, L. 31: please report goodness of fit / error measures of the site specific rating curves.

Response:

We have amended and the sentence now reads. 'The levels for the Glenamong and Srahrevagh rivers were converted to volume of discharge per second (m3 s-1) using site specific rating curves: Glenamong R2 = 0.98, Srahrevagh - R2 = 0.9677 (Marine Institute unpublished data).'

Comment:

p. 7, L. 6-9: How does this analysis relate to the study aims? Also, I'd appreciate a motivation for the choice of the statistical tests. Was the A Wilcoxon Signed Rank Test used to account for the nestedness of the Black and Srahrevagh rivers?

Response:

This section now reads. 'This analysis was conducted to ascertain if there was significant statistical difference between river colour in each catchment. A Mann-Whitney U test was used to test for statistical differences between colour concentrations in the Glenamong and Black and the Glenamong and Srahrevagh rivers. As the Black and Srahrevagh rivers are in the same river system, a non-parametric Wilcoxon Signed Rank Test was used to test for statistical differences between their colour concentrations.'

Comment:

p. 7, L. 11: please clarify "Loess"

Response:

The following sentence was added: 'Loess (locally weighted smoothing) regression is a nonparametric technique that uses local weighted regression to fit a smooth curve through points in a scatter plot.'

Comment:

p. 7, L.11: please give a reference for the R program used (move it up from p. 7, L. 29).

Response:

Reference has been moved.

Comment:

p. 7, L. 23- p. 8 L. 19: How were the GAMM models reduced to find the optimum model with three smoothers?

Response:

The optimum models were found by an iterative process described in Zuur et al. (2009). Explanatory variables were initially tested for any evidence of collinearity. No variables that were collinear were used in the same model. The resultant GAMs were then first tested for any breach of the assumption of equal variance. If required, a variance structure was added, with the optimum structure selected based on AIC values. Any non-significant variables were then removed from the model. As a next step, tests were carried out for any breach of the assumption of independence. If found in the models, various autocorrelation structures were tested and the optimum structure was added, again based on the AIC value. This model was the final 'optimum' model.

Comment:

p. 7, L. 25: is the mgcv package an R package? Please indicate.

Response:

This sentence now reads. 'To identify the main explanatory drivers of colour in the rivers, general additive mixed models (GAMM) with cubic smoothing regression splines and Gaussian distributions were developed using the mgcv package in R (Wood, 2006).'

Comment:

Equation 1: use italics consistently, i.e. even for $T_{lt}$

Response:

Equation has been corrected in the manuscript

Comment:

p. 8, L. 6: What is the motivation to include wind speed, radiation and humidity in the model? Background / hypotheses for testing these variables are not given in the introduction.

Response:

Higher wind speeds are positively correlated with higher air temperatures at this site, due to warm westerly air masses. Since decomposition of peat and therefore production of DOC is sensitive to temperature, there was potential for a relationship. However, none was found.

Comment:

p. 8, L. 12: How did you use NAO in the statistical analysis? As explanatory variables? Please clarify.

Response:

This sentence now reads. 'Both daily and monthly means of the NAO index were downloaded from the National Oceanic and Atmospheric Administration (NOAA, 2017) and used as explanatory variables in the statistical analysis.'

Comment:

p. 8, L. 13: please clarify the abbreviation "SMD"

Response:

We note that the abbreviation is explained where soil moisture deficit is first mentioned in the manuscript, page 8 line 12: 'Hydrological explanatory variables included river discharge (m3 s-1), soil moisture deficit (SMD) (mm day-1) and actual and potential evapotranspiration (mm day -1).'

Comment:

p. 8, L. 21: what exactly is meant with "to further examine the linkages between each of the explanatory drivers"? I would expect many readers to be unfamiliar with the crosswavelet transform analysis (including myself) and would appreciate a clarification, in simple words, what the analysis does.

Response:

We agree with the reviewer and have rewritten as follows. 'A cross-wavelet transform analysis was carried out to further examine the trends and periodicities in colour concentrations with the explanatory drivers of colour in the rivers. Cross-wavelet transform analysis can be used as a method of examining pairs of time series that may be expected to be linked in some way. Continuous wavelet transforms from pairs of time series are used to construct the cross wavelet transforms, revealing their common power and relative phase in time-frequency space. In particular, the analysis examines whether regions in time frequency space with large common power have a consistent phase relationship, suggesting causality between the time series pairs (Grinstead et al, 2004).'

Comment:

p. 8, L. 27: Please give more details on the Monte Carlo methods used!

Response:

The Monte Carlo methods used are imbedded within the R programme / package algorithms, and we have decided to remove the reference to Monte Carlo methods in the manuscript. The sentence now reads: 'A cross-wavelet power spectrum was calculated from the cross wavelet transform results in order to estimate the covariance between each pair of time series as a function of frequency and the statistical significance was also as part of the analysis.

Comment:

p. 8, L. 32: were the residuals of the linear regression between DOC and color homoscedastic?

Response:

Variance was checked for the residuals from each GAMM using the gam.check function in mgcv. This indicated that the residuals were homoscedastic and therefore a variance structure was not added to the model

Comment:

p. 9, L. 1: add details on the location of the vendor of the DOC analyzer.

Response:

Sievers 5310C Total Organic Carbon analyser (Sievers Instruments, Inc. sievers.instruments.wts@suez.com)

Comment:

p. 9. L 4-5: What is meant with "mean load" and "annual load"? is it the annual mean load referred to here?

Response:

We have corrected the sentence to now read. 'The annual load was calculated by multiplying the calculated stream discharge volume for each week by the weekly DOC concentration and summing the totals for each year'.

Comment:

p. 9, L. 19: which time period is referred to here? 1995 to ... ?

Response:

The sentence has been clarified and now reads:

'A comparison of monthly precipitation values during the six year study period with monthly precipitation for the previous 15 years (1995 to 2010). . .'

Comment:

p. 9, L. 28: is the "top 10%" the 90% percentile?

Response:

This sentence now reads. 'Values greater than the 90% percentile of discharge for the Black river were $> 4.47$ m3 s$-1$ while those less than the 10% percentile were $< 0.26$ m3 s$-1$ (Figure 2B).'

Comment:

p. 11, L. 2: what does "edf" stand for?

Response:

Sentence now reads: 'The smoother explaining the relationship between soil temperature and colour was linear in the model (estimated degrees of freedom (edf) = 1) and positive, indicating that colour increases with increasing temperature.'

Comment:

p. 11, L. 12-25: I very much appreciate the sensitivity analysis, investigating the performance of the GAMM model depending on whether SMD or discharge is included!

Response:

We felt we should include this for completeness.

Comment:

p. 11, L. 26-31: How strong was the correlation between NAO, soil temperature and SMD? It comes somehow through in Fig. 5, but some metric describing this correlation might add further valuable context to the relatively low contribution of NAO in addition to the effect of soil temperature and SMD.

Response:

We draw the reviewers attention to the F statistic in Table 2 which gives an indication of the relative contribution of each explanatory variable to each model. For the NAO this would be approximately 4% for the GAMM for the Black, and 2.5% for the Glenamong and for the Srahrevagh

Comment:

p. 13, L. 8: To my understanding, DeFries and Eshleman (2004) only discuss forestry effects on hydrology, not DOC export. Please refer some of the many papers that show increased DOC loads in response to forest clear-felling (e.g. Nieminen 2004, Silva Fennica 38(2); Schelker et al. 2012.

Response:

This sentence now reads: 'Forestry is also known to influence DOC release from soils and it has been observed that both afforestation and forest clear-felling result in increased DOC concentrations and that these increases may continue for several years after the initial event (Cummins and Farrell, 2003; Schelker et al. 2012).'

Comment:

p. 13, L. 10: how about replacing "goes some way in" by "may help"

Response:

Agreed, and the sentence now reads: 'Spatial analysis, comparing the extent of peat soils in the study catchments, the length of streams intersecting the peat, slope analysis and CORINE land cover in each sub-catchment, may help in explaining the higher levels of colour found in the Srahrevagh.'

Comment:

p. 13, L. 12: How much is known about the forestry intensity in the catchments? Is the forest clear-cut? In Table 1, only the areal proportion of forest (based on CORINE data) is given, but this does not imply that the forests are managed. Is this the same information that is given in Fig. 1 (symbol code "forestry")? More information on forestry is needed to support the statement that forest clearcutting could explain differences in DOC loads across catchments.

Response:

We agree that more information should be provided and therefore the text has been amended to read: 'All of the coniferous forestry in the catchment is owned by the semi-state company Coillte or managed by private forestry companies. These areas of forestry are intensively managed, and when the timber is harvested the forests are generally clear-cut.'

Comment:

p.13, L. 27-30: Would the interaction with water table fluctuations imply that correlations between soil temperature and water color is low at time scales Âń 1 year (as apparent from Fig. 6)? If so, I'd suggest to refer to results shown in Fig. 6 here. Response:

For clarity, the cross-wavelet analysis showed that soil temperature and water colour

were almost exactly in-phase, with a significant positive correlation. This indicated that when soil temperature was higher/lower, stream water colour was also higher/lower. This cross-wavelet analysis also indicated that there was very little correlation between these two variables at time scales other than the annual time scale across all six years. Based on these results, we concluded that the relationship between water colour and soil temperature indicated by the GAMM operated mainly at the annual time scale. We have stated this in the text as follows: 'For soil temperature, the width of the orientation at the annual time step was relatively consistent with phase arrows that all pointed right, i.e. there was a positive correlation between soil temperature and river colour that was consistent at the annual scale'

Comment:

p. 14, L. 4: to test this, would it be possible to run the cross-wavelet analysis for time scales longer than 1 year?

Response:

To clarify, this analysis included all data over the full 6 year study period, and assessed correlations at all time steps up to 64. There were not sufficient data to analyse at time periods longer than 64 weeks.

Comment:

p. 15, L. 18: the term "weather" is maybe not optimal here. How about "low pressure systems" or "cyclones", etc...?

Response:

This sentence now reads. 'During the construction of the GAMMs, both the weekly and monthly NAO index values were tested in the analysis. The models using the weekly data consistently explained more of the deviance in the model. This most likely reflects the proximity of the site to the Atlantic coast and the time frame over which weather systems associated with the NAO pressure difference generally reached the

study location.'

Figures and Tables:

Comment:

Fig. 1: referee 1 asked what did the green-blueish areas mean in the figure and pointed out that this color code was not explained in the figure legend.

Response: These areas have been removed from the figure.

Comment:

Fig. 1: is the weather station the "Newport" met station? Please indicate.

Response: Yes, this has been clarified in the figure.

Comment:

Fig. 1: please explain the red dot in the map of Ireland.

Response: This is the study site, it has been clarified in the legend.

Comment:

Fig. 2: explain the abbreviation "SMD" in the figure caption.

Response: This has been explained in the figure caption.

Comment:

Fig. 3: improve the resolution and contrast of the figure

Response: The resolution and contrast of the figure has been improved.

Comment:

Fig. 3, caption: add "water" in front of "color".

Response: This has been done

Comment:

Fig. 3, caption: the letters "B", "C" and "D" appear in the wrong position. Please correct.

Response: The figures have been re-positioned.

Comment:

Fig. 4, caption: explain the meaning of "s" shown on the y-axis scales.

Response: this has been clarified.

Comment:

Fig. 4, caption: explain the abbreviation "SMD"

Response: this has been clarified.

Comment:

Fig. 5: I cannot find an explanation in the methods section on how this analysis was done. Please clarify. Details on the trend analysis of water color is given, but not for the environmental driver variables.

Response: this has been clarified in the methods section.

Comment:

Fig. 5: the figure appears stretched along the x-axis in my version. Please modify.

Response: This has been resolved in the figure.

Comment:

Fig. 5, caption: please indicate the time scale of the trends shown. Is it weekly averages?

Response: They are weekly averages, this has been clarified.

[Figure]

Comment:

Fig. 5, caption: explain the abbreviation "SMD"

Response: this has been clarified.

Comment:

Fig. 6: What do the line graphs on top of the contour plots indicate? Also, the tick marks along the axes of these line graphs are hardly visible. Please increase font size and add axis labels.

Response: The line graphs show the two time-series datasets being analysed, however given the small size of the contour plots, it will be quite difficult to create legible titles and axis labels, we propose to remove the line graphs in fig.6 in the final manuscript.

Comment:

Fig. 6: Which depth does soil temperature refer to?

Response:

Soil temperature is at 1 m depth. This has been added to the figure caption.

Comment:

Fig. 6, caption: explain the abbreviation "SMD" Response: this has been clarified.

Comment:

Fig. 6, caption: What are edge effects and what is the cone of influence? Please explain here or in the methods section.

Response: These have been clarified in the caption as follows: 'Pink regions on either end indicate areas where the analysis is unreliable as there is no data before and after the study period.'

Comment:

Table 1: is the climatological data given here recorded at the Newport met station? Please indicate.

Response: this has been clarified in the caption.

Comment:

Table 1: "stream length" can differ a lot depending on how / at which spatial resolution it is mapped. how is "stream length" defined? What is the smallest system (e.g. in terms of upslope contributing catchment area) considered here?

Response: Stream length was sourced from a national coverage of streams and lakes that are available from the Irish Environmental Protection Agency and available as a shape file for download and use in GIS software.

Comment:

Table 2: please explain the abbreviations "edf" and "Ref.df". These values are identical. Why?

Response: We agree with the reviewer: edf has been explained in the Table caption and Ref.df has been removed from the table.

Comment:

Table 3, caption: maybe mention that Lough Feeagh is the lake shown in Fig. 1, or indicate the lake name in Fig. 1?

Response: We have stated that Lough Feeagh is the main lake in Fig. 1 in the caption.

Comment:

Table 3: it was not immediately clear to me that the seasonal DOC loads given in the lower part of the table are linked to the years listed in the table header. Maybe explain that in the figure caption?

Response: this has been clarified in the Table caption.
Comment:

Table 3: please explain abbreviations D, J, F,

Response: this has been clarified in the Table caption.

Comment:

Table 3: Shouldn't the sum of the seasonal DOC loads equal the annual DOC loads? This is at least not the case here. Why?

Response: The seasonal DOC loads in winter run from December to February, part of this season is in the previous year (December) therefore the seasonal and annual don't tally.

Comment:

Use consistent abbreviations ("Fig.") for "figure". Some figures are not referred to in the text in the same order as they appear in the figure section. For example, Fig. 3 is referred to before Fig. 2 is referred to.

Response: this has been corrected in the manuscript.

Comment:

P. 2, L. 21: delete one of the "each": done. P. 4, L. 10: "trend" should be plural to be consistent with "changes" mentioned before: done. P. 5, L. 26: delete one of the "spatially": done. P. 6, L. 29 replace the first "," by "and", and remove the second ",": done. P. 7, L. 16: delete one of the "for": done. P. 8, L. 23: use "were" instead of "are": done. P. 9, L. 1: add a full stop between "rivers" and "doc": done. P. 15 , L. 27: remove the full stop between "that" and "both": done. Fig. 2: remove "(" after "standardized precipitation index" at the y-axis label of the uppermost panel.: done. p. 24, L. 4: there is a digit missing in "201" Table 1 caption: "sub-catchmen" is missing a "t": done.

Response:

[Figure]

The above typographical errors were also corrected.

References: Clark, J.M., Lane, S.N., Chapman, P.J. and Adamson, J.K., 2008. Link between DOC in near surface peat and stream water in an upland catchment. Science of the Total Environment, 404(2-3), pp.308-315

Ritson, J.P., Brazier, R.E., Graham, N.J., Freeman, C., Templeton, M.R. and Clark, J.M., 2017. The effect of drought on dissolved organic carbon (DOC) release from peatland soil and vegetation sources.

Ryder, E., de Eyto, E., Dillane, M., Poole, R., and Jennings, E.: Identifying the role of environmental drivers in organic carbon export from a forested peat catchment, Sci. Total Environ., 490: 28–36, 2014.